# Schrödinger Bridge Matching for Tree-Structured Costs and Entropic Wasserstein Barycentres

**Samuel Howard**
Department of Statistics
University of Oxford

**Peter Potaptchik**
Department of Statistics
University of Oxford

**George Deligiannidis**
Department of Statistics
University of Oxford

## Abstract

Recent advances in flow-based generative modelling have provided scalable methods for computing the Schrödinger Bridge (SB) between distributions, a dynamic form of entropy-regularised Optimal Transport (OT) for the quadratic cost. The successful Iterative Markovian Fitting (IMF) procedure solves the SB problem via sequential bridge-matching steps, presenting an elegant and practical approach with many favourable properties over the more traditional Iterative Proportional Fitting (IPF) procedure. Beyond the standard setting, optimal transport can be generalised to the multi-marginal case in which the objective is to minimise a cost defined over several marginal distributions. Of particular importance are costs defined over a tree structure, from which Wasserstein barycentres can be recovered as a special case. In this work, we extend the IMF procedure to solve for the tree-structured SB problem. Our resulting algorithm inherits the many advantages of IMF over IPF approaches in the tree-based setting. In the case of Wasserstein barycentres, our approach can be viewed as extending the widely used fixed-point approach to use flow-based entropic OT solvers, while requiring only simple bridge-matching steps at each iteration. Our code is available at `https://github.com/samuel-howard/Tree_SB_Matching_Barycentres`.

## 1 Introduction

Transporting mass between two distributions is a ubiquitous problem with numerous applications in machine learning and beyond. Optimal Transport (OT) (Santambrogio, 2015; Peyré and Cuturi, 2019) provides a principled approach for such problems, by seeking to minimise the total cost of transportation according to a chosen cost function. Since the introduction of Sinkhorn's algorithm (Cuturi, 2013) and more recent neural approaches (Makkuva et al., 2020), computational OT has seen great success across many domains such as biology (Schiebinger et al., 2019; Bunne et al., 2023), and extensively in machine learning (Genevay et al., 2018; Cuturi et al., 2019; Corenflos et al., 2021). Recently, ideas from the powerful flow-based approaches that have revolutionised generative modelling (Song et al., 2021; Peluchetti, 2022; Lipman et al., 2023; Liu et al., 2023; Albergo and Vanden-Eijnden, 2023) have been leveraged to solve the entropy-regularised *dynamic* OT problem, known as the Schrödinger Bridge (SB). Such approaches provide significant scalability advantages, enabling approximation of OT maps between high-dimensional continuous datasets such as image data. Early flow-based SB solvers were based on the classical Iterative Proportional Fitting (IPF) scheme (De Bortoli et al., 2021; Vargas et al., 2021; Chen et al., 2022), but such methods have since been superseded by those based on the Iterative Markovian Fitting (IMF) scheme (Shi et al., 2023; Peluchetti, 2023) due to its many superior properties.

Beyond the standard OT problem between two marginals, multi-marginal OT aims to find a joint coupling over *multiple* marginals while minimising the total cost. Tree-structured costs are often

---

Corresponding author: `howard@stats.ox.ac.uk`

39th Conference on Neural Information Processing Systems (NeurIPS 2025).

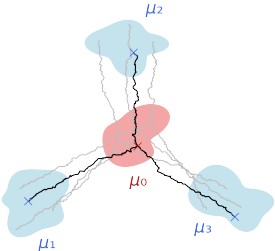
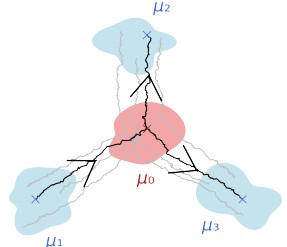
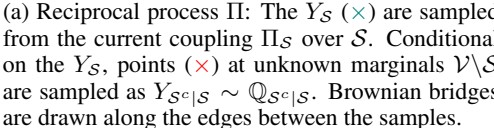

(a) Reciprocal process $\Pi$: The $Y_{\mathcal{S}}$ ($\times$) are sampled from the current coupling $\Pi_{\mathcal{S}}$ over $\mathcal{S}$. Conditional on the $Y_{\mathcal{S}}$, points ($\times$) at unknown marginals $\mathcal{V}\backslash\mathcal{S}$ are sampled as $Y_{\mathcal{S}^c|\mathcal{S}} \sim \mathbb{Q}_{\mathcal{S}^c|\mathcal{S}}$. Brownian bridges are drawn along the edges between the samples.

(b) Markovianised process $\mathbb{P}$: Vector fields are trained by bridge-matching along each edge. Samples ($\times$) from the next coupling $\Pi_{\mathcal{S}}$ are obtained by simulating the resulting SDEs along the tree structure, started at one of the known marginals.

Figure 1: The two stages of TreeIMF. On this tree, the marginals at the leaf vertices $\mathcal{S}$ (blue) are fixed. The marginal at the vertex in $\mathcal{V}\backslash\mathcal{S}$ (red) is not fixed, and can change during the procedure.

considered (Haasler et al., 2021), as they frequently arise in applications while also allowing for improved scalability by leveraging the tree structure. Of particular significance are star-shaped trees, as these correspond to the prominent Wasserstein barycentre problem (Agueh and Carlier, 2011). The Wasserstein barycentre provides a natural notion of 'average' for probability distributions, and is widely studied due to its importance in applications, including in Bayesian learning (Srivastava et al., 2018), clustering (Ye et al., 2017), and representation learning (Singh et al., 2020) to name a few. Computing barycentres is notoriously challenging (Altschuler and Boix-Adserà, 2022). Many successful approaches approximate the solution with a finite set of points, but these in-sample methods struggle to scale well as dimensionality increases. Alternative methods aim to provide continuous approximations (Li et al., 2020) using neural parameterisations, and often require multi-level optimisation procedures (Fan et al., 2021; Korotin et al., 2022). Of particular relevance to our work is the diffusion-based approach of Noble et al. (2023) which extends the IPF-based Diffusion Schrödinger Bridge (DSB) method of De Bortoli et al. (2021) to the tree-based SB setting, and is (to our knowledge) currently the only neural ODE/SDE approach for barycentre computation.

One of the most successful and elegant approaches for Wasserstein-2 barycentre computation is the iterative fixed-point approach, which involves iteratively updating a candidate barycentre $\nu$ by solving for the OT map from $\nu$ to each marginal $\mu_i$ and updating $\nu$ according to the induced coupling. This approach was popularised by the seminal work of Álvarez-Esteban et al. (2016) and has formed the basis of many algorithmic developments, including in machine learning (Korotin et al., 2022). It has strong performance, and has been observed to converge quickly in only a few iterations (Lindheim, 2023). However, the procedure requires solving a complete OT problem to each marginal at each iteration, which is expensive as solving even a single OT problem can be challenging.

**Contributions** In this work, we extend the IMF procedure to the tree-based SB problem (Haasler et al., 2021). Our TreeDSBM algorithm provides an IMF counterpart to the TreeDSB method from Noble et al. (2023), closing a clear gap in the existing literature (see Table 1) and translating the many benefits of IMF over IPF to the tree setting. For the specific case of Wasserstein barycentre computation, our algorithm can be viewed as extending the commonly used fixed-point approaches to the case of flow-based entropic OT solvers. In particular, we show that the iterations of the IMF scheme and the fixed-point barycentre solvers can be elegantly combined into a single iterative procedure, yielding a fixed-point-style algorithm that requires only inexpensive bridge-matching at each iteration. We demonstrate significantly improved empirical performance over TreeDSB, and show that flow-based barycentre solvers can offer competitive performance against existing algorithms for continuous Wasserstein-2 barycentre computation.

Table 1: Positioning of our TreeDSBM algorithm in the literature.

|  | **SB** | **TreeSB** (Haasler et al., 2021) |
|---|---|---|
| **IPF** | DSB (De Bortoli et al., 2021) | TreeDSB (Noble et al., 2023) |
| **IMF** | DSBM (Shi et al., 2023; Peluchetti, 2023) | TreeDSBM (**Ours**) |

## 2 Background

### 2.1 Schrödinger bridges, optimal transport, and Wasserstein barycentres

We begin by reviewing the standard Schrödinger Bridge (SB) problem between two distributions, and its relation to Optimal Transport (OT). For notation used in the paper, see Appendix A. Given an initial and final measure $\mu_0$ and $\mu_T$ of a population, the SB problem aims to identify the most likely intermediate dynamics of the population under the assumption that the movement is driven by a stochastic reference process $\mathbb{Q}$. The resulting *dynamic* SB problem is defined as

$$\mathbb{P}^{SB} = \underset{\mathbb{P} \in P(\mathcal{C})}{\arg\min} \{\mathrm{KL}(\mathbb{P} \| \mathbb{Q}) \mid \mathbb{P}_0 = \mu_0, \mathbb{P}_T = \mu_T\}. \tag{1}$$

The *static* SB problem instead considers only the coupling over the endpoints,

$$\Pi_{0,T}^{SB} = \underset{\Pi_{0,T} \in \mathcal{P}(\mathbb{R}^d \times \mathbb{R}^d)}{\arg\min} \{\mathrm{KL}(\Pi_{0,T} \| \mathbb{Q}_{0,T}) \mid \Pi_0 = \mu_0, \Pi_T = \mu_T\}. \tag{2}$$

Under mild assumptions, the two problems are equivalent; the dynamic SB solution can be expressed as a mixture of bridges over the static SB coupling, $\mathbb{P}^{SB} = \Pi_{0,T}^{SB} \mathbb{Q}_{\cdot|0,T}$ (Léonard, 2013).

**Connection to quadratic OT** The reference path measure $\mathbb{Q}$ is usually considered to be that of a Brownian motion $(\sigma B_t)_{t \in [0,T]}$. In this case, the static SB problem can be rewritten as

$$\Pi_{0,T}^{SB} = \underset{\Pi \in \mathcal{P}(\mathbb{R}^d \times \mathbb{R}^d)}{\arg\min} \{\mathbb{E}\|X_0 - X_T\|_2^2 - 2\sigma^2 T \mathrm{H}(\Pi) \mid \Pi_0 = \mu_0, \Pi_T = \mu_T\}, \tag{3}$$

which is the entropy-regularised OT problem for the quadratic ground-cost $c(x_0, x_T) = \frac{1}{2}\|x_0 - x_T\|^2$ and $\varepsilon = \sigma^2 T$ (see the Appendix for an overview of OT). In the sequel, we assume that the reference process is a Brownian motion $(\sigma B_t)_{t \in [0,T]}$.

**Wasserstein barycentres** A key motivation for the tree-structured setting that we will consider is the important Wasserstein barycentre problem (Agueh and Carlier, 2011), which provides a natural notion of 'average' for probability distributions. Given $\ell$ measures $(\mu_1, ..., \mu_\ell)$ and weights $(\lambda_1, ..., \lambda_\ell)$ summing to 1, the Wasserstein-2 barycentre is defined as

$$\nu^* = \arg\min_\nu \sum_i \lambda_i W_2^2(\mu_i, \nu), \tag{4}$$

where $W_2^2(\mu_i, \nu)$ denotes the optimal transport cost between $\mu_i$ and $\nu$ for the quadratic ground-cost.

The Wasserstein barycentre problem is notoriously difficult to solve, as it involves several OT sub-problems for which one of the marginals can change in the optimisation. An elegant approach leverages the fixed-point property $x = \sum_i \lambda_i T_i(x)$ which holds at the solution (under mild assumptions), where each $T_i$ is the OT map from $\nu$ to $\mu_i$ (Agueh and Carlier, 2011; Álvarez-Esteban et al., 2016). Intuitively, this states that 'each point in the support of the barycentre is the average of the corresponding points in the marginals'. This property has motivated an iterative fixed-point approach for barycentre computation, which involves iteratively updating a candidate barycentre $\nu$ by solving for the OT maps $T_i$ and constructing the next iterate as $\bar{T}\#\nu$ using the pushforward map $\bar{T}(x) = \sum_i \lambda_i T_i(x)$. Such methods have been shown to be highly successful in practice (Álvarez-Esteban et al., 2016; Korotin et al., 2022; Lindheim, 2023; Tanguy et al., 2024).

### 2.2 Iterative Markovian Fitting

We now outline recent flow-based generative modelling approaches for solving for the SB problem, which will form the basis for our approach. The SB solution $\mathbb{P}^{SB}$ can be characterised as the *unique* path measure that is both Markov, and a mixture of bridges $\mathbb{P}^{SB} = \mathbb{P}_{0,T}^{SB} \mathbb{Q}_{\cdot|0,T}$, that has correct marginals $\mathbb{P}_0^{SB} = \mu_0, \mathbb{P}_T^{SB} = \mu_T$ (Léonard, 2013). This property motivates the Iterative Markovian Fitting (IMF) procedure (Shi et al., 2023; Peluchetti, 2023), which solves for the SB solution by alternately projecting between Markovian processes and processes with the correct bridges. We refer to Shi et al. (2023) for full details of the IMF procedure, but recall here the basic presentation. We recall the following definitions.

**Definition 2.1** (**Reciprocal class, Reciprocal projection**). *A path measure $\Pi \in \mathcal{P}(\mathcal{C})$ is in the reciprocal class $\mathcal{R}(\mathbb{Q})$ of $\mathbb{Q}$ if it is a mixture of bridges of $\mathbb{Q}$ conditional on their values at the endpoints, $\Pi = \Pi_{0,T} \mathbb{Q}_{\cdot|0,T}$. For a path measure $\mathbb{P} \in \mathcal{P}(\mathcal{C})$, the reciprocal projection is defined to be the mixture of bridges according to its induced coupling, $\mathrm{proj}_{\mathcal{R}(\mathbb{Q})}(\mathbb{P}) = \mathbb{P}_{0,T} \mathbb{Q}_{\cdot|0,T}$.*

**Definition 2.2** (**Markovian class, Markovian projection**). *Let $\mathcal{M}$ denote the set of Markovian path measures associated to a diffusion of the form $\mathrm{d}X_t = v(t, X_t)\mathrm{d}t + \sigma_t \mathrm{d}B_t$, with $v, \sigma$ locally Lipschitz. For reference process $(\sigma B_t)_{t \in [0,T]}$, the Markovian projection of a measure $\mathbb{P} \in \mathcal{R}(\mathbb{Q})$ is defined to be (when well-defined) the path measure associated to the SDE*

$$\mathrm{d}X_t = \left[ \frac{\mathbb{E}_{\mathbb{P}_{T|t}}[X_T|X_t] - X_t}{T - t} \right]\mathrm{d}t + \sigma \mathrm{d}B_t, \qquad X_0 \sim \mu_0. \tag{5}$$

It can be shown that, under mild conditions, these definitions coincide with the following minimisation problems over path measures (Shi et al., 2023),

$$\mathrm{proj}_{\mathcal{R}(\mathbb{Q})}(\mathbb{P}) = \underset{\Pi \in \mathcal{R}(\mathbb{Q})}{\arg\min}\, \mathrm{KL}(\mathbb{P}\|\Pi), \qquad \mathrm{proj}_{\mathcal{M}}(\Pi) = \underset{\mathbb{M} \in \mathcal{M}}{\arg\min}\, \mathrm{KL}(\Pi\|\mathbb{M}). \tag{6}$$

The IMF iterations are defined below, and converge to a unique fixed point which is the SB solution.

$$\mathbb{P}^{2n+1} = \mathrm{proj}_{\mathcal{M}}(\mathbb{P}^{2n}), \qquad \mathbb{P}^{2n+2} = \mathrm{proj}_{\mathcal{R}(\mathbb{Q})}(\mathbb{P}^{2n+1}). \tag{7}$$

**Training via bridge-matching**   The IMF procedure requires learning the vector field corresponding the Markovian projections $\mathrm{proj}_{\mathcal{M}}(\Pi)$. This is done by training a neural network $v_\theta$ with a *bridge-matching* loss objective (Peluchetti, 2022), for which the drift in (5) is the optimum,

$$\mathcal{L}(\theta) = \int_0^T \underset{\substack{(X_0, X_T) \sim \Pi_{0,T} \\ X_t \sim \mathbb{Q}(\cdot | X_0, X_T)}}{\mathbb{E}} \| v_\theta(X_t, t) - \frac{X_T - X_t}{T - t} \|^2 \mathrm{d}t. \tag{8}$$

**Comparison to Iterative Proportional Fitting**   Traditionally, the standard way to solve the SB problem is via the Iterative Proportional Fitting (IPF) procedure (Fortet, 1940; Kullback, 1968; Rüschendorf, 1995), which minimises the KL divergence between subsequent iterations while alternating the endpoint measure that is fixed.

$$\mathbb{P}^{2n+1} = \underset{\mathbb{P}:\mathbb{P}_T = \mu_T}{\arg\min}\, \mathrm{KL}(\mathbb{P}\|\mathbb{P}^{2n}), \qquad \mathbb{P}^{2n+2} = \underset{\mathbb{P}:\mathbb{P}_0 = \mu_0}{\arg\min}\, \mathrm{KL}(\mathbb{P}\|\mathbb{P}^{2n+1}). \tag{9}$$

This method was implemented using diffusion model-based approaches in De Bortoli et al. (2021), giving the Diffusion Schrödinger Bridge (DSB) method (see also Vargas et al. (2021), Chen et al. (2022)). While allowing for arbitrary reference measures $\mathbb{Q}$, this approach suffers from several shortfalls, such as only preserving both marginals at convergence (in comparison to IMF, which preserves both marginals at each iteration), as well as expensive trajectory caching, and 'forgetting' of the original reference measure (Fernandes et al., 2022). As a result, IPF approaches have largely been superseded by IMF approaches which avoid these issues.

### 2.3   Tree-structured Schrödinger bridge

In this work, we consider the tree-structured Schrodinger Bridge problem (Haasler et al., 2021). In particular, we extend the IMF procedure to the *dynamic* Tree SB setting of Noble et al. (2023).

**Stochastic processes on the tree**   In order to explain the dynamic Tree SB problem, we first need to define stochastic processes on a tree. We will consider trees $\mathcal{T} = (\mathcal{V}, \mathcal{E}, \ell)$ with vertex set $\mathcal{V}$ and edge set $\mathcal{E}$, where $\ell : \mathcal{E} \to \mathbb{R}^{>0}$ is an edge-length function. One can extend the tree to a uniquely arcwise-connected metric space in the natural way; the arc connecting the two endpoints of an edge $e$ is identified with a line segment of corresponding length $T^e = \ell(e)$, and they are connected according to the graph structure. Via a slight abuse of notation we will also denote this metric space as $\mathcal{T}$. As such, we can define the space $C(\mathcal{T}, \mathbb{R}^d)$ of continuous paths from $\mathcal{T}$ into $\mathbb{R}^d$, and we let $\mathcal{P}(\mathcal{C}_{\mathcal{T}}) := \mathcal{P}(C(\mathcal{T}, \mathbb{R}^d))$ denote the space of probability measures over such paths. We will present the methodology according to a *directed* tree $\mathcal{T}_r$ rooted at a vertex $r$, for which we can choose an ordered edge set $\mathcal{E}_r$ corresponding to a depth-first traversal (though we will often omit the dependence on $r$ in the notation). While this presentation may appear somewhat complex, the construction is quite natural; for an illustration of processes on a tree-structure, see Figures 1 and 4.

We will consider running SDEs along each edge according to the ordering $\mathcal{E}_r$ in the directed tree structure. We sample $X_r \sim \mathbb{P}_r$ and then sequentially simulate SDEs along each edge as we traverse the directed tree. For each edge $e = (u, v) \in \mathcal{E}_r$, we run an SDE $\mathrm{d}X_t^e = v^e(t, X_t^e)\mathrm{d}t + \sigma_t^e \mathrm{d}B_t^e$ for time $t \in [0, T^e]$ initialised at $X_0^e = X_u$, and after simulation we let $X_v = X_{T^e}^e$. Such stochastic processes induce a path measure $\mathbb{P} \in \mathcal{P}(\mathcal{C}_{\mathcal{T}})$ over the whole tree. Note that when the tree is a branch with 2 vertices and 1 edge, this recovers the standard case described earlier.

**Tree-structured Schrödinger bridges**  Now that we have constructed stochastic processes on the tree, the tree-structured Schrödinger Bridge problem is defined analogously to the standard case. However, now the marginals may be fixed only at a *subset* of vertices $\mathcal{S} \subset \mathcal{V}$ (see Figure 1). For a reference measure $\mathbb{Q} \in \mathcal{P}(\mathcal{C}_{\mathcal{T}})$, the *dynamic* and *static* TreeSB problems are defined respectively as

$$\mathbb{P}^{SB} = \underset{\mathbb{P} \in \mathcal{P}(\mathcal{C}_{\mathcal{T}})}{\arg\min} \{ \mathrm{KL}(\mathbb{P}\|\mathbb{Q}) \mid \mathbb{P}_i = \mu_i \ \forall i \in \mathcal{S} \}, \qquad\qquad (\mathrm{TreeSB_{dyn}})$$

$$\Pi_{\mathcal{V}}^{SB} = \underset{\Pi \in \mathcal{P}((\mathbb{R}^d)^{|\mathcal{V}|})}{\arg\min} \{ \mathrm{KL}(\Pi\|\mathbb{Q}_{\mathcal{V}}) \mid \Pi_i = \mu_i \ \forall i \in \mathcal{S} \}. \qquad (\mathrm{TreeSB_{stat}})$$

As in the standard case, using the chain rule for KL divergence gives (under mild conditions) equivalence of the two problems; the dynamic SB solution can be expressed as a mixture of bridges over the static SB coupling, $\mathbb{P}^{SB} = \Pi_{\mathcal{V}}^{SB} \mathbb{Q}_{\cdot|\mathcal{V}}$. In this work, we aim to solve the $\mathrm{TreeSB_{dyn}}$ problem.

**Connection to multi-marginal optimal transport**  In the remainder of the work, we will consider the reference measure $\mathbb{Q}$ to be associated with running Brownian motions $(\sigma B_t^e)_{t \in [0, T^e]}$ along each edge $e$. The induced reference coupling $\mathbb{Q}_{\mathcal{V}}$ over the vertices is characterised by having independent Gaussian edge increments $Y_u - Y_v \sim \mathcal{N}(0, \sigma^2 T^{(u,v)})$. By evaluating the KL term, the static TreeSB problem can therefore be rewritten as

$$\Pi_{\mathcal{V}}^{SB} = \underset{\Pi \in P((\mathbb{R}^d)^{|\mathcal{V}|})}{\arg\min} \left\{ \mathbb{E}_{X_{\mathcal{V}} \sim \Pi}\Big[ \sum_{(u,v) \in \mathcal{E}} \frac{1}{T^{(u,v)}} \|X_u - X_v\|_2^2 \Big] - 2\sigma^2 \mathrm{H}(\Pi)|\Pi_i = \mu_i \ \forall i \in \mathcal{S} \right\}. \quad (10)$$

This is precisely an entropy-regularised multi-marginal OT problem with a tree-structured quadratic cost function $c(x_{\mathcal{V}}) = \sum_{(u,v) \in \mathcal{E}} \frac{1}{T^{(u,v)}} \|x_u - x_v\|_2^2$ and entropy-regularisation $\varepsilon = 2\sigma^2$ (see Appendix A for an overview). Note that each weight is the reciprocal of the corresponding edge length.

**Connection to Wasserstein barycentres**  There are many ways to define *entropy-regularised* Wasserstein barycentres (see Appendix A for an overview). The work of Chizat (2023) unifies several of these, combining $\varepsilon$-regularised OT costs with an entropic penalty term on the barycentre weighted by $\tau$. Noble et al. (2023) show that the following doubly-regularised barycentre problem

$$\nu^* = \arg\min_{\nu} \left\{ \sum_i \lambda_i W_{2, \varepsilon/\lambda_i}^2(\mu_i, \nu) + (\ell - 1)\varepsilon \mathrm{H}(\nu) \right\}, \qquad\qquad (11)$$

is recovered from the TreeSB setting by considering a star-shaped tree with fixed marginals on the leaves and cost function $c(x_{0:\ell}) = \sum_{i=1}^{\ell} \lambda_i \|x_0 - x_i\|_2^2$.

## 3  Iterative Markovian Fitting for tree-structured costs

In this work, we generalise the IMF procedure to the tree-based setting. Recall that standard IMF proceeds by iteratively Markovianising a mixture of bridges, where each bridge is a stochastic process conditioned on endpoints $x_0$ and $x_1$, and the mixing coupling $\Pi_{0,1}$ is over the endpoint values $x_0, x_1$. Similarly, we consider Markovianising mixtures of bridges, but instead the mixing coupling is a joint distribution $\Pi_{\mathcal{S}}$ over values at the observed vertices $\mathcal{S}$, and the bridging processes are conditioned on these values. In the following, we justify that the IMF procedure extends to this setting. We then explain how the bridging and Markovianisation can be performed in practice.

### 3.1  Markovian and reciprocal processes on the tree

Recall that the IMF procedure is motivated by the characterisation of the SB as the unique Markov measure that is a mixture of its bridges. Our first contribution is to extend this characterisation to the tree-based setting. We begin with the following definitions, akin to those in the standard case.

**Definition 3.1 (Markov class).** *Let $\mathcal{M}_{\mathcal{T}}$ denote the set of Markov path measures on the tree $\mathcal{T}$, defined via a diffusion along each edge of the form $dX_t^e = v^e(t, X_t)dt + \sigma_t^e dB_t^e$, with $v^e, \sigma^e$ locally Lipschitz.*

**Definition 3.2 (Reciprocal class).** *$\Pi \in \mathcal{P}(\mathcal{C}_{\mathcal{T}})$ is in the reciprocal class $\mathcal{R}_{\mathcal{S}}(\mathbb{Q})$ of $\mathbb{Q}$ for observed vertices $\mathcal{S}$ if $\Pi = \Pi_{\mathcal{S}}\mathbb{Q}_{\cdot|\mathcal{S}}$. That is, $\Pi$ is a mixture of bridges of $\mathbb{Q}$ conditioned on the values at $\mathcal{S}$.*

Observe that compared to the corresponding Definition 2.1 in standard IMF, the bridges are now conditioned on the values at vertices in $\mathcal{S}$, rather than on only the endpoints of a single edge. We now state the following result characterising the TreeSB solution.

**Theorem 3.1** (**TreeSB characterisation**). *Under mild assumptions (in the Brownian case, namely that $\int \|x\|^2 d\mu_i(x) < \infty$ and $\mathrm{H}(\mu_i) < \infty$ for each $i \in \mathcal{S}$), there exists a unique solution to the dynamic TreeSB problem* (TreeSB$_{\mathrm{dyn}}$). *The solution is the unique process $\mathbb{P}$ that is both Markov and in $\mathcal{R}_\mathcal{S}(\mathbb{Q})$ with correct marginals $\mathbb{P}_i = \mu_i$ for $i \in \mathcal{S}$.*

## 3.2 Iterative Markovian Fitting on the tree

Theorem 3.1 suggests that an analogous Iterative Markovian Fitting procedure could be used to solve the dynamic TreeSB problem; in this section, we show that this is indeed the case. The constructions and proofs proceed in much the same way as in Shi et al. (2023), but crucially rely on the following simple decomposition of the KL divergence according to the tree structure, which applies for measures in $\mathcal{M}_\mathcal{T}$ and $\mathcal{R}_\mathcal{S}(\mathbb{Q})$.

**Lemma 3.2** (**KL decomposition along tree**). *Take path measures $\mathbb{P}, \tilde{\mathbb{P}}$ that share a marginal at the root, $\mathbb{P}_r = \tilde{\mathbb{P}}_r$, and factorise along the tree. Under mild assumptions, we have the KL decomposition*

$$\mathrm{KL}(\mathbb{P}\|\tilde{\mathbb{P}}) = \sum_{(u,v) \in \mathcal{E}_r} \mathbb{E}_{X_u \sim \mathbb{P}_u} \Big[ \mathrm{KL}(\mathbb{P}^{(u,v)}(\cdot|X_u)\|\tilde{\mathbb{P}}^{(u,v)}(\cdot|X_u)) \Big]. \tag{12}$$

To extend the IMF procedure to the TreeSB problem, we require to extend the definitions and propositions from the standard case to the tree-based setting. The definition of the reciprocal class projection remains much the same as in the standard case, the difference being that now the bridging processes are conditioned on their values at vertices in $\mathcal{S}$.

**Definition 3.3** (**Reciprocal projection**). *For a process $\mathbb{P} \in \mathcal{P}(\mathcal{C}_\mathcal{T})$, the reciprocal projection is defined to be the induced mixture of bridges $\Pi^* = \mathrm{proj}_{\mathcal{R}_\mathcal{S}(\mathbb{Q})}(\mathbb{P}) = \mathbb{P}_\mathcal{S}\mathbb{Q}_{\cdot|\mathcal{S}}$.*

**Proposition 3.3.** *For $\mathbb{P} \in \mathcal{P}(\mathcal{C}_\mathcal{T})$, the reciprocal projection $\Pi^* = \mathrm{proj}_{\mathcal{R}_\mathcal{S}(\mathbb{Q})}(\mathbb{P})$ solves the minimisation problem $\Pi^* = \arg\min_{\Pi \in \mathcal{R}_\mathcal{S}(\mathbb{Q})} \mathrm{KL}(\mathbb{P}\|\Pi)$.*

The Markovian projection requires some alterations to account for the tree structure.

**Definition 3.4** (**Markovian projection**). *For $\Pi \in \mathcal{R}_\mathcal{S}(\mathbb{Q})$, the Markovian projection onto $\mathbb{M}$, denoted $\mathbb{M}^* = \mathrm{proj}_\mathcal{M}(\Pi)$, is defined as*

$$\mathbb{M}^* = \prod_\mathcal{T} \mathrm{proj}_{\mathcal{M}_e}(\Pi_e). \tag{13}$$

$\Pi_e$ *denotes the restriction of $\Pi$ to edge $e$, $\mathrm{proj}_{\mathcal{M}_e}(\Pi_e)$ denotes its Markovian projection (Definition 2.2), and $\prod_\mathcal{T}$ denotes the composition of the resulting measures according to the tree structure.*

**Proposition 3.4.** *Under mild assumptions, the Markovian projection $\mathbb{M}^* = \mathrm{proj}_\mathcal{M}(\Pi)$ solves the minimisation problem $\mathbb{M}^* = \arg\min_{\mathbb{M} \in \mathcal{M}_\mathcal{T}} \mathrm{KL}(\Pi\|\mathbb{M})$.*

The TreeIMF procedure is then defined via the usual iterations, started from an initialisation $\mathbb{P}^0 \in \mathcal{R}_\mathcal{S}(\mathbb{Q})$ such that $\mathbb{P}_i^0 = \mu_i$ for $i \in \mathcal{S}$:

$$\mathbb{P}^{2n+1} = \mathrm{proj}_{\mathcal{M}_\mathcal{T}}(\mathbb{P}^{2n}), \quad \mathbb{P}^{2n+2} = \mathrm{proj}_{\mathcal{R}_\mathcal{S}(\mathbb{Q})}(\mathbb{P}^{2n+1}). \tag{14}$$

Note that this procedure recovers standard IMF when the tree has a bridge structure (2 vertices, 1 edge), just as TreeDSB recovers DSB in this setting.

We now state the following result, which shows convergence of the IMF iterates to the TreeSB solution. This resembles Theorem 8 in Shi et al. (2023), and the proof proceeds in largely the same way using the appropriate modifications and in particular leveraging the KL decomposition in Lemma 3.2. We defer the details to Appendix B.

**Theorem 3.5** (**Convergence of TreeIMF**). *Under mild conditions, the* TreeSB$_{\mathrm{dyn}}$ *solution $\mathbb{P}^*$ is the unique fixed point of the TreeIMF iterates $\mathbb{P}^n$, and we have $\lim_{n \to \infty} \mathrm{KL}(\mathbb{P}^n\|\mathbb{P}) = 0$.*

### 3.3 Implementation

We have established the convergence of the TreeIMF procedure to the TreeSB solution. We now explain how the procedure can be implemented in practice.

**Constructing the bridge processes**  Recall that the reciprocal projection relies on constructing bridges of the reference process $Y$ associated to $\mathbb{Q}$, conditioned on the values at vertices in $\mathcal{S}$. Due to the tree structure and the Brownian dynamics of $\mathbb{Q}$, such bridges can be constructed by first sampling the values of $Y_{\mathcal{S}^c|\mathcal{S}} \sim \mathbb{Q}_{\mathcal{S}^c|\mathcal{S}}$ at the unseen vertices in $\mathcal{S}^c = \mathcal{V}\backslash\mathcal{S}$, and then sampling Brownian bridges along each edge $(u,v) \in \mathcal{E}$ between $Y_u$ and $Y_v$.

Note in particular that the conditional coupling $\mathbb{Q}_{\mathcal{S}^c|\mathcal{S}}$ is tractable because the static coupling $\mathbb{Q}_{\mathcal{V}}$ is a multivariate Gaussian. Specifically, $\mathbb{Q}_{\mathcal{V}}$ is characterised via independent Gaussian edge increments $Y_u - Y_v \sim \mathcal{N}(0, \sigma^2 T^e)$, so $\mathbb{Q}_{\mathcal{V}}(Y) \propto \exp(-\sum_{(u,v)\in\mathcal{E}} \frac{1}{2\sigma^2 T^e}\|Y_u - Y_v\|^2) \propto \exp(-\frac{1}{2}Y^\top L Y)$ where $L$ is the precision matrix given by

$$L_{u,v} = \begin{cases} \sum_{e\in\mathcal{E}(u)} \frac{1}{\sigma^2 T^{(u,v)}} & \text{if } u = v, \text{ where } \mathcal{E}(u) \text{ is set of edges incident to } u, \\ -\frac{1}{\sigma^2 T^{(u,v)}}, & \text{if } (u,v) \in \mathcal{E} \\ 0 & \text{otherwise.} \end{cases}$$

Using the formula for conditional multivariate Gaussians in terms of the precision matrix, we have $Y_{\mathcal{S}^c}|Y_{\mathcal{S}} = y_{\mathcal{S}} \sim \mathcal{N}(\tilde{\mu}, \tilde{\Sigma})$, using the block matrix expressions $\tilde{\mu} = -(L_{\mathcal{S}^c\mathcal{S}^c})^{-1}L_{\mathcal{S}^c\mathcal{S}}y_{\mathcal{S}}$ and $\tilde{\Sigma} = (L_{\mathcal{S}^c\mathcal{S}^c})^{-1}$. Based on the structure of the tree, we can therefore obtain a sample of the bridging process by: (1) calculating this conditional joint distribution $\mathbb{Q}_{\mathcal{S}^c|\mathcal{S}}$ over the unseen vertices, (2) sampling $Y_{\mathcal{S}^c|\mathcal{S}}$ according this distribution, and then (3) drawing Brownian bridges between $Y_u$ and $Y_v$ along each edge in the tree.

**Performing the Markovian projection**  The above construction allows us to construct samples from a reciprocal process $\Pi$. Recall from Definition 3.4 that the Markovian projection then proceeds by performing individual Markovian projections along each edge. We therefore maintain a neurally-parameterised drift function $v_{\theta_e}$ for each edge $e = (u,v)$, each of which is trained according to the bridge-matching loss

$$\mathcal{L}(\theta_e) = \int_0^{T^e} \mathbb{E}_{\substack{(X_u,X_v)\sim\Pi_{(u,v)} \\ X_t \sim \mathbb{Q}^e_{\cdot|X_u,X_v}}} \|\frac{X_v - X_t}{T^e - t} - v_{\theta_e}(X_t, t)\|^2 \mathrm{d}t. \tag{15}$$

**Bidirectional training**  After training the vector fields, let $\mathbb{M}$ denote the path measure of the resulting Markov process on $\mathcal{T}$. Constructing the next reciprocal process requires samples from $\mathbb{M}$, which can be obtained by running the learned diffusions from the root node $r \in \mathcal{S}$ along the tree. Note that while in theory we have $\mathbb{M}_i = \mu_i$ for each $i \in \mathcal{S}$, errors will accumulate in practice. We therefore follow Shi et al. (2023) and learn both forward and backward diffusion processes along each edge, which give equivalent representations of the path measures via the corresponding time-reversals (see Appendix C for details). This enables simulation along the tree from any vertex in $\mathcal{S}$ to obtain samples from the coupling, helping to mitigate errors accumulating in the marginals. Note also that all edges train independently, so can be learned in parallel for faster computation.

In keeping with previous naming conventions, we call the proposed algorithm Tree Diffusion Schrödinger Bridge Matching (TreeDSBM). The method is summarised in Algorithm 1 with full implementation details in Appendix C, and we provide an illustration of the two-step procedure in Figure 1. The algorithm can be initialised at any coupling $\Pi_{\mathcal{S}}^0$ over $\mathcal{S}$ with correct marginals; a standard choice would be the independent coupling $\Pi_{\mathcal{S}}^0 = \bigotimes_{i\in\mathcal{S}} \mu_i$. We provide a simplified and more explicit version of the algorithm when used for barycentre computation in Appendix C.

---

**Algorithm 1:** TreeDSBM

**Input:** Initial coupling $\Pi_{\mathcal{S}}^0$ (e.g. independent), number of iterations $N$.

Let $\Pi^0 = \Pi_{\mathcal{S}}^0 \mathbb{Q}_{\cdot|\mathcal{S}}$;

**for** $n \in \{0, \ldots, N-1\}$ **do**

    Learn $2|\mathcal{E}|$ vector fields using (15) with $\Pi = \Pi^n$, to obtain Markovian process $\mathbb{M}^{n+1}$;

    Simulate $\mathbb{M}^{n+1}$ from a chosen root $r \in \mathcal{S}$ to obtain samples from $\mathbb{M}_{\mathcal{S}}^{n+1}$;

    Let $\Pi^{n+1} = \mathbb{M}_{\mathcal{S}}^{n+1} \mathbb{Q}_{\cdot|\mathcal{S}}$ using obtained samples from $\mathbb{M}_{\mathcal{S}}^{n+1}$;

**end**

---

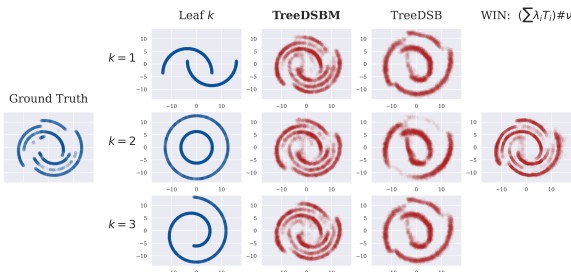

Figure 2: Comparison of the learned barycentre for TreeDSBM (6 IMF iterations) against TreeDSB (50 IPF iterations) and WIN. TreeDSB and TreeDSBM samples are generated from each leaf vertex $k$, and for WIN we plot samples using the weighted-pushforward expression for the barycentre. Also displayed is a close approximation to the ground-truth, using the in-sample method of Cuturi and Doucet (2014).

Table 2: Sinkhorn divergence to the 'ground-truth' barycentre, for barycentre samples generated from each leaf vertex $k$. The Sinkhorn divergence is computed with entropy regularisation 0.01, using 5000 generated samples and 1500 points approximating the ground-truth (mean±std, over 5 runs).

| Method | $k = 0$ | $k = 1$ | $k = 2$ |
|---|---|---|---|
| TreeDSBM (6 IMF) | 1.14 $\pm\,0.07$ | 1.05 $\pm\,0.07$ | 1.08 $\pm\,0.11$ |
| TreeDSB (50 IPF) | 2.35 | 4.04 | 2.35 |
| WIN $(\sum_i \lambda_i T_i)\#\nu$ | | 1.17 | |

## 3.4 Connection to fixed-point Wasserstein barycentre algorithms

Recall that the unregularised Wasserstein-2 barycentre can be computed using the iterative fixed-point method, where each iteration proceeds by pushing-forward the current iterate through the map $\sum_i \lambda_i T_i$. In our setting, the barycentre problem corresponds to a star-shaped tree with fixed marginals on the leaves. In this case, the bridges of $\mathbb{Q}$ are conditioned on each of the leaf vertices and the only unknown point is $Y_0$ at the centre vertex, so the conditional distribution $\mathbb{Q}_{\mathcal{S}^c|\mathcal{S}}$ simplifies to

$$Y_0 \sim \mathcal{N}\Big(\sum \lambda_i Y_i, \sigma^2 Id\Big).$$

Our method can therefore be viewed as a natural counterpart of fixed-point methods for barycentre computation, adapted to the case of flow-based entropic OT solvers. While a naive approach might consider using IMF to solve each OT sub-problem in the fixed-point scheme, resulting in nested iterations, our approach shows that the IMF and fixed-point iterations can in fact be elegantly combined into a single iterative procedure, along with a well-understood theoretical grounding. In particular, the expensive OT map computations required for the fixed-point procedure can instead be switched out for inexpensive bridge-matching procedures. Each iteration is therefore cheap, and empirically we found the TreeDSBM algorithm to retain the fast convergence property of the fixed-point procedure in terms of the number of iterations required.

## 4 Experiments

**Synthetic 2$d$ barycentre** We first examine the performance of TreeDSBM in a low-dimensional synthetic example, using the experimental setup of Noble et al. (2023). We compute the $(\frac{1}{3}, \frac{1}{3}, \frac{1}{3})$-barycentre of a moon, spiral, and circle dataset with $\varepsilon = 0.1$ (recall $\sigma = \sqrt{\frac{\varepsilon}{2}}$). We compare TreeDSBM ran for 6 IMF iterations against TreeDSB ran for 50 IPF iterations (using checkpoints provided by Noble et al. (2023)). In Figure 2, we plot the obtained samples from both methods, and for comparison also display the barycentre obtained using the in-sample method from Cuturi and Doucet (2014) to give a close approximation to the ground-truth. To quantitatively assess performance, we report the Sinkhorn divergence (Genevay et al., 2018) relative to this 'ground-truth' in Table 2. The results show that TreeDSBM *significantly* improves over TreeDSB in this setting, approximating the barycentre to a high degree of accuracy and at a much lower computational cost (with good convergence after only a few IMF iterations). It is able to successfully capture the complex nature of the barycentre in this challenging example (previously suggested as a potential limitation of dynamic solvers in Noble et al. (2023)), and the barycentres generated from different vertices $k$ exhibit improved consistency. For details of the experimental setup, see Appendix D.1.

We additionally report results for the iterative WIN method (Korotin et al., 2022). We see that TreeDSBM performs competitively with this strong baseline for continuous Wasserstein-2 solvers. The results reported for WIN are for the combined map $(\sum_i \lambda_i T_i)\#\nu$, as in our experiments the barycentre generator $\nu = G\#\rho$ was unable to fit the true barycentre accurately, nor were the maps $T_i^{-1}\#\mu_i$. Additionally, we applied the W2CB (Korotin et al., 2021) and NOTWB (Kolesov et al.,

2024a) algorithms to this example, but were unable find hyperparameters to make the algorithms to converge to the correct solution. We hypothesise that the neural maps may struggle to model the discontinuous transports well, and also that this challenging example may result in difficult loss landscapes. We found TreeDSBM to exhibit stable training despite this challenging problem setting, and to also be the fastest of the algorithms to converge to the solution. We provide a runtime analysis, along with further results and discussion, in Appendix D.1.

**MNIST 2,4,6 barycentre**  We also compare performance of TreeDSBM with TreeDSB on a higher dimensional image dataset, computing the $(\frac{1}{3}, \frac{1}{3}, \frac{1}{3})$-barycentre between MNIST digits 2, 4, and 6 (LeCun et al., 2010). In this setting, TreeDSB is reported in Noble et al. (2023) to exhibit training instability for low entropy regularisation $\varepsilon$, causing it to struggle to match the marginal measures at the vertices. In contrast, the IMF approach of TreeDSBM ensures matching of these marginals, thus allowing the use of much smaller regularisation values. In fact, using a large regularisation for TreeDSBM would limit sample quality, because of the noise added when sampling from $\mathbb{Q}_{\mathcal{S}^c|\mathcal{S}}$ in the reciprocal process construction. We there-

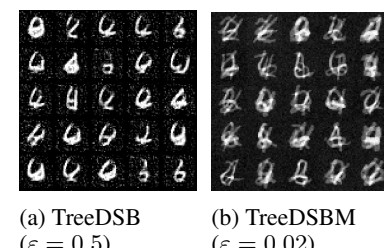

(a) TreeDSB ($\varepsilon = 0.5$)    (b) TreeDSBM ($\varepsilon = 0.02$)

Figure 3: Samples from the 2,4,6 MNIST barycentre.

fore use $\varepsilon = 0.02$ for TreeDSBM with 4 IMF iterations, and display a visual comparison with results reported in Noble et al. (2023) in Section 4. While it is difficult to validate the accuracy of such solutions, the obtained samples from TreeDSBM appear to display a greater resemblance to state-of-the-art barycentre methods for similar problems (see e.g. Kolesov et al. (2024a)), while again requiring significantly less computational cost than TreeDSB. We remark that the TreeDSBM samples in Figure 3b contain a small amount of noise, which is to be expected as we are solving for an entropy-regularised problem. In Appendix D.2 we also display samples with the noise reduced or removed, along with full experimental details and additional results.

**Subset posterior aggregation**  The previous experiments have shown TreeDSBM to improve over its IPF counterpart TreeDSB. We now provide a more detailed comparison with existing strongly-performing methods for continuous Wasserstein-2 barycentre estimation, namely WIN (Korotin et al., 2022), W2CB (Korotin et al., 2021), and NOTWB (Kolesov et al., 2024a). As a real-world applications experiment, we compare the algorithms in the subset posterior aggregation setting, a standard experiment in the barycentre literature (Staib et al., 2017; Li et al., 2020; Fan et al., 2021; Korotin et al., 2021). The barycentre of subset posteriors is known to be close to the true posterior (Srivastava et al., 2018), and as such it can be efficient to compute posteriors on only subsets of datasets before aggregating them using a barycentre algorithm. We consider the experimental setup and dataset used in Korotin et al. (2021) (the same dataset was also used previously in Li et al. (2020) and Fan et al. (2021)), which uses Poisson and negative-binomial regressions on a bike-rental dataset (Fanaee-T, 2013), and report results for the $\mathrm{BW}_2^2{-}\mathrm{UVP}$ metric. We ran TreeDSBM for 4 IMF iterations with $\varepsilon = 0.001$; for full experimental details, see Appendix D.3.

From Table 3, we see that all methods perform strongly. Given that we do not have perfect access to the ground truth barycentre, it is difficult to conclusively say which performs best, but we see that TreeDSBM certainly performs competitively with these state-of-the-art approaches. Moreover, we found TreeDSBM to display fast training—both TreeDSBM and NOTWB obtained good results after only around 3 minutes of training, while W2CB took approximately 10 minutes and WIN took around 45 minutes. For more details regarding runtimes, see Appendix D.3.

**Higher-dimensional Gaussian experiments**  We also report results for computing the barycentre of Gaussian distributions, in increasingly high dimensions. This is a standard experiment in the literature, because in this setting the ground-truth barycentre is also Gaussian and the parameters can be calculated accurately using a fixed point method (Álvarez-Esteban et al., 2016).

We follow the experimental setup previously used in (Korotin et al., 2021, 2022; Kolesov et al., 2024a), in which 3 Gaussian distributions and its ground-truth $(\frac{1}{3}, \frac{1}{3}, \frac{1}{3})$-barycentre are randomly generated, for each dimension in $\{64, 96, 128\}$. We report results for the $\mathrm{BW}_2^2{-}\mathrm{UVP}$ and $\mathrm{L}^2{-}\mathrm{UVP}$ metrics. We see that all the methods again perform well in this setting. For the $\mathrm{BW}_2^2{-}\mathrm{UVP}$ metric, W2CB and NOTWB appear to have a slight edge, but TreeDSBM is only slightly higher and is comparable with results for WIN. For the $L^2{-}\mathrm{UVP}$ metric, results for TreeDSBM are higher than for W2CB and NOTWB, though the results are still low and are again comparable with WIN.

Table 3: $\mathrm{BW}_2^2-\mathrm{UVP}, \%$ for different algorithms on the subset posterior aggregation experiment in Korotin et al. (2021), evaluated using 100,000 samples.

|  | WIN | W2CB | NOTWB | TreeDSBM (Ours) |
|---|---|---|---|---|
| ↓ Poisson | 0.014 | 0.026 | 0.023 | 0.008 |
| ↓ Negative Binomial | 0.009 | 0.024 | 0.018 | 0.012 |

Table 4: $L^2-\mathrm{UVP}, \%$ and $\mathrm{BW}_2^2-\mathrm{UVP}, \%$ for different algorithms on the high-dimensional Gaussian experiment, evaluated using 100,000 samples.

|  |  | WIN | W2CB | NOTWB | TreeDSBM (Ours) |
|---|---|---|---|---|---|
| ↓ $\mathrm{BW}_2^2-\mathrm{UVP}, \%$ | $d = 64$ | 0.20 | 0.04 | 0.08 | 0.14 |
|  | $d = 96$ | 0.30 | 0.07 | 0.10 | 0.15 |
|  | $d = 128$ | 0.38 | 0.12 | 0.14 | 0.27 |
| ↓ $L^2-\mathrm{UVP}, \%$ | $d = 64$ | 0.96 | 0.17 | 0.10 | 1.18 |
|  | $d = 96$ | 1.20 | 0.20 | 0.10 | 1.13 |
|  | $d = 128$ | 1.46 | 0.25 | 0.13 | 1.23 |

## 5  Discussion

**Related work**   Beyond the IPF and IMF approaches, other approaches for the SB problem include adversarial solvers (Kim et al., 2024; Gushchin et al., 2023), parameterisation via potentials for Gaussian mixtures (Korotin et al., 2024; Gushchin et al., 2024), and variational approaches (Deng et al., 2024). For Wasserstein barycentre computation, standard approaches ensure tractability by representing the solution with a finite set of points (either updating only weightings (Benamou et al., 2015; Solomon et al., 2015; Cuturi and Peyré, 2016; Staib et al., 2017; Dvurechenskii et al., 2018), or also the positions of the points in the support (Rabin et al., 2012; Cuturi and Doucet, 2014; Claici et al., 2018; Luise et al., 2019)). Such approaches are effective in lower dimensions but scale poorly, cannot be used to generate new samples, and do not capture the true continuous nature of the barycentre. To address these limitations, recent work has focused on learning continuous approximations. Li et al. (2020) optimise for the regularised dual potentials for general costs, while others parameterise Wasserstein-2 potentials with convex neural architectures using adversarial losses (Fan et al., 2021) or additional cycle-consistency regularisation (Korotin et al., 2021). Korotin et al. (2022) leverage the fixed-point property in Álvarez-Esteban et al. (2016). We also highlight a recent line of works of Kolesov et al. (2024b,a) which consider bi-level adversarial approaches for general cost functions. Our approach utilises non-adversarial bridge-matching loss objectives, which provide stable training but requires multiple sequential IMF iterations. Finally, we emphasise that our approach tackles the tree-structured SB problem (Haasler et al., 2021; Noble et al., 2023), and thus is applicable beyond only barycentre problems (we consider a toy example in Appendix E.3).

**Limitations**   Our method shares the same limitations as other flow-based SB solvers. It is restricted to quadratic cost functions for OT, and introduces an entropic bias. Inference is expensive in comparison to methods that use a single function evaluation, and our method is not sampling-free, requiring simulations of the current learned processes at each iteration. Future advances in the flow and SB literatures can aid in addressing these limitations. We also provide an additional experiment in Appendix E that highlights a possible limitation of our method in scenarios where there is a simple shared structure between the known marginals and the barycentre.

**Conclusion and future work**   We have extended the IMF procedure to the tree-structure SB problem, providing a scalable flow-based approach that in particular can be used for entropic Wasserstein barycentre computation. Our TreeDSBM algorithm displays improved performance over its IPF counterpart TreeDSB, and demonstrates that flow-based approaches for barycentre estimation can offer a compelling alternative to established continuous Wasserstein-2 barycentre algorithms. Future directions can investigate improved architectures and implementation techniques inspired by progress in the flow-matching literature, as well as extensions to other data modalities.

## Acknowledgements

SH is supported by the EPSRC CDT in Modern Statistics and Statistical Machine Learning [grant number EP/S023151/1]. PP is supported by the EPSRC CDT in Modern Statistics and Statistical Machine Learning [EP/S023151/1], a Google PhD Fellowship, and an NSERC Postgraduate Scholarship (PGS D). GD was supported by the Engineering and Physical Sciences Research Council [grant number EP/Y018273/1]. The authors would like to thank James Thornton for helpful discussions.

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

# Appendix

The Appendix is structured in the following way. In Appendix A, we define the notation used and provide further background information regarding the problems discussed in the paper. In Appendix B, we provide the proofs of the results stated in the main body. In Appendix C, we describe in more detail how to implement the TreeDSBM algorithm, and also discuss extensions of the methodology for improving convergence speed. In Appendix D, we give a full overview of the experimental setups in the main body, and include further discussions of the findings. In Appendix E we provide additional experiments that could not be included in the main body due to space constraints. Finally, Appendix F includes the licenses for assets used in this work.

## A  Background

### A.1  Notation

We denote by $\mathcal{T}$ a tree with an edge length function, $\mathcal{T} = (\mathcal{V}, \mathcal{E}, \ell)$, or its associated metric space via associating edges $e$ with the line segment of length $T^e = \ell(e)$ (see Section 2.3). We let $\mathcal{P}(\mathcal{C}) = \mathcal{P}(C([0,T], \mathbb{R}^d))$ and $\mathcal{P}(\mathcal{C}_\mathcal{T}) := \mathcal{P}(C(\mathcal{T}, \mathbb{R}^d))$ denote the space of path measures on the interval $[0,T]$ and $\mathcal{T}$ respectively. For the interval $[0,T]$, $\mathcal{M}$ denotes the set of Markov measures, and $\mathcal{R}(\mathbb{Q})$ denotes the reciprocal class for a reference measure $\mathbb{Q} \in \mathcal{M}$ (see Definitions 2.1 and 2.2). For a path measure $\mathbb{Q} \in \mathcal{P}(\mathcal{C})$, $\mathbb{Q}_t$ denotes the marginal distribution at time $t$, and $\mathbb{Q}_{\cdot|0,T}$ denotes the diffusion bridge conditional on values at $0, T$. We use $\Pi = \Pi_{0,T} \mathbb{Q}_{\cdot|0,T} \in \mathcal{P}(\mathcal{C})$ to denote mixtures of bridges $\Pi(\cdot) = \int_{\mathbb{R}^d \times \mathbb{R}^d} \mathbb{Q}_{\cdot|0,T}(\cdot|x_0, x_T) \Pi_{0,T}(\mathrm{d}x_0, \mathrm{d}x_T)$. Similarly on the tree $\mathcal{T}$, $\mathcal{M}_\mathcal{T}$ denotes the set of Markov measures, and $\mathcal{R}_\mathcal{S}(\mathbb{Q})$ denotes the reciprocal class for a reference measure $\mathbb{Q} \in \mathcal{M}_\mathcal{T}$ and observed vertices $\mathcal{S} \subset \mathcal{V}$ (see Definitions 3.1 and 3.2). For a path measure $\mathbb{Q} \in \mathcal{P}(\mathcal{C}_\mathcal{T})$, $\mathbb{Q}_v$ denotes the marginal at vertex $v \in \mathcal{V}$, $\mathbb{Q}_\mathcal{V}$ denotes the induced joint distribution over a set of vertices $\mathcal{V}$, and $\mathbb{Q}_{\cdot|\mathcal{S}}$ denotes the bridge process conditional on values at vertices $\mathcal{S}$. We denote mixtures of bridges as $\Pi = \Pi_\mathcal{S} \mathbb{Q}_{\cdot|\mathcal{S}} \in \mathcal{P}(\mathcal{C}_\mathcal{T})$, defined similarly to above. Finally, $\mathrm{KL}(\pi \| \tilde{\pi})$ denotes the Kullback-Leibler divergence between measures $\pi, \tilde{\pi}$, and $\mathrm{H}(\pi)$ denotes the differential entropy.

### A.2  Optimal transport

We here provide a brief overview of optimal transport; for more details we refer to Santambrogio (2015), Peyré and Cuturi (2019).

**Monge**  For two measures $\mu_0$ and $\mu_1$, the original OT formulation given by Monge (1781) aims to find a map $T^* : \mathbb{R}^d \to \mathbb{R}^d$ pushing $\mu_0$ onto $\mu_1$, that is $\mu_1 = T^* \# \mu_0$, while minimising the total cost of transportation according to a given cost function $c : \mathbb{R}^d \times \mathbb{R}^d \to \mathbb{R}$,

$$\min_{T: T\#\mu_0=\mu_1} \int_{\mathbb{R}^d} c(x, T(x)) \mathrm{d}\mu_0(x). \tag{16}$$

**Kantorovich**  While intuitive, the Monge formulation of OT is restrictive in that it does not permit splitting of mass in the transportation. Instead, it is common to consider the more general Kantorovich formulation (Kantorovich, 1942), which searches over joint distributions $\pi \in \mathcal{P}(\mathbb{R}^d \times \mathbb{R}^d)$ with the correct marginals,

$$\min_{\pi: \pi_0=\mu_0, \pi_1=\mu_1} \iint_{\mathbb{R}^d \times \mathbb{R}^d} c(x, y) \mathrm{d}\pi(x, y). \tag{17}$$

**Entropic regularisation**  The entropy-regularised OT problem adds an entropic penalty term to the Kantorovich objective, which smooths the resulting transport plan.

$$\min_{\pi: \pi_0=\mu_0, \pi_1=\mu_1} \left\{ \iint_{\mathbb{R}^d \times \mathbb{R}^d} c(x, y) \mathrm{d}\pi(x, y) - \varepsilon \mathrm{H}(\pi) \right\}.$$

The resulting problem has many desirable properties; it enables differentiability with respect to the inputs, relaxes constraints on the corresponding potentials (Genevay et al., 2016), and for discrete measures enables efficient computation using Sinkhorn's algorithm (Cuturi, 2013).

### A.3 Multi-marginal optimal transport

One can generalise the above optimal transport problem to the case of multiple marginal distributions. For a cost function $c : (\mathbb{R}^d)^{\ell+1} \to \mathbb{R}$ and an 'observed' set $\mathcal{S} \subset \{0, ..., \ell\}$, multi-marginal optimal transport (mmOT) searches over joint distributions $\pi \in \mathcal{P}((\mathbb{R}^d)^{\ell+1})$ matching the prescribed marginals on $\mathcal{S}$ to minimise the objective,

$$\min_{\pi : \pi_i = \mu_i \ \forall i \in \mathcal{S}} \int_{(\mathbb{R}^d)^{\ell+1}} c(x_{0:\ell}) \mathrm{d}\pi(x_{0:\ell}). \tag{18}$$

As in the standard case, one can define an entropy-regularised version of the mmOT problem,

$$\min_{\pi : \pi_i = \mu_i \ \forall i \in \mathcal{S}} \left\{ \int_{(\mathbb{R}^d)^{\ell+1}} c(x_{0:\ell}) \mathrm{d}\pi(x_{0:\ell}) - \varepsilon \mathrm{H}(\pi) \right\}. \tag{19}$$

**Multi-marginal Schrödinger bridges**  We emphasise here the distinction between the multi-marginal OT and corresponding SB problem that we consider in this work, compared to the multi-marginal SB problem that has been considered recently in works such as Chen et al. (2019), Chen et al. (2023), and Shen et al. (2025) to name a few. These lines of works aim to find processes that evolve in time while fitting to known marginals at several different timepoints, and are motivated by inferring population dynamics from snapshot data. We rather consider costs defined according to a tree structure, in which the marginals at some of the vertices may not be known (as considered in Haasler et al. (2021) and Noble et al. (2023)), with a particular focus on applications to Wasserstein barycentre problems.

### A.4 Wasserstein barycentres

Given $\ell$ measures $(\mu_1, ..., \mu_\ell)$ and weights $(\lambda_1, ..., \lambda_\ell)$ summing to 1, the Wasserstein-2 barycentre (Agueh and Carlier, 2011) is defined as

$$\nu^* = \arg\min_{\nu} \sum_i \lambda_i W_2^2(\mu_i, \nu), \tag{20}$$

where $W_2^2(\mu_i, \nu)$ denotes the minimum attained by the OT solution in (17) for quadratic cost function $c(x, y) = \frac{1}{2}\|x - y\|_2^2$. Observe that this can be cast as an mmOT problem by using a star-shaped cost function $c(x_{0:\ell}) = \sum_{i=1}^{\ell} \lambda_i \|x_0 - x_i\|_2^2$, where the marginals are prescribed on the leaf nodes $\mathcal{S} = \{1, ..., \ell\}$ and the centre vertex 0 is an unobserved measure (which at the solution is the barycentre). The Wasserstein barycentre is widely studied due to its importance in applications, including in Bayesian learning (Srivastava et al., 2018), clustering (Ye et al., 2017), and representation learning (Singh et al., 2020) to name a few.

**Types of regularised barycentre**  There are many different ways to define an entropy-regularised formulation of the Wasserstein barycentre problem. Some formulations add *inner* regularisation, which replaces the Wasserstein distances with an entropy-regularised version, while others also consider *outer* regularisation in which an entropic penalty on the barycentre is added. The $(\varepsilon, \tau)$-doubly-regularised barycentre of Chizat (2023) unifies many of these problems, and aims to minimise the objective

$$\min_{\nu} \left\{ \sum_i \lambda_i W_{2,\varepsilon}^2(\mu_i, \nu) - \tau H(\nu) \right\}, \tag{21}$$

where $W_{2,\varepsilon}^2(\mu_i, \nu) = \min_{\pi : \pi_0 = \mu_i, \pi_1 = \nu} \left\{ \iint \frac{1}{2}\|x - y\|^2 \mathrm{d}\pi(x, y) + \varepsilon \mathrm{KL}(\pi \| \mu \otimes \nu) \right\}$. We refer to Chizat (2023) for more details regarding the different types of entropy-regularised Wasserstein barycentres.

**Tree-structured costs**  The case of tree-structured costs for multi-marginal optimal transport is often specifically studied in the literature (Haasler et al., 2021; Noble et al., 2023), as it recovers Wasserstein barycentres as a special case as described above, and also arises in the Wasserstein propagation problem (Solomon et al., 2014, 2015). In the discrete setting, Haasler et al. (2021) show that the tree structure can be leveraged to design an efficient Sinkhorn-based algorithm.

**The metric space $\mathcal{T}$**  Recall from Section 2.3 that we identify a tree $\mathcal{T}$ (with vertices, edge set, and length function $(\mathcal{V}, \mathcal{E}, \ell)$) with a metric space, by associating each edge $e$ with the interval $[0, \ell(e)]$, which are connected according to the tree structure. The same construction is used in Noble et al. (2023) for defining the dynamic TreeSB problem that we study in this work. For a rigorous description of such constructions, see for example Hambly and Lyons (2008), Bolin et al. (2024).

# B  Proofs

In this section, we give proofs of the results stated in the main paper. We first provide the proof of the TreeSB characterisation in Theorem 3.1. We will follow the presentation and proof techniques of Léonard (2013), making the appropriate changes to extend to the tree-structured case. We then prove the convergence of the TreeIMF procedure stated in Theorem 3.1, following the presentation and proof techniques of Shi et al. (2023).

## B.1  Existence and uniqueness of TreeSB solution

We first provide results pertaining to the existence and uniqueness of the TreeSB solution. Note that while we follow the presentation of Léonard (2013), one could instead rely on the existence results in Noble et al. (2023).

Let us define another static minimisation problem only over observed vertices in $\mathcal{S}$ as

$$\Pi_{\mathcal{S}}^{SB} = \underset{\Pi \in \mathcal{P}((\mathbb{R}^d)^{|\mathcal{S}|})}{\arg\min} \{\mathrm{KL}(\Pi\|\mathbb{Q}_{\mathcal{S}}) \mid \Pi_i = \mu_i \ \forall i \in \mathcal{S}\}. \qquad (\text{TreeSB}_{\text{stat}}^{\mathcal{S}})$$

We note here that the reference measures that we consider in the various SB problems are unbounded, as we consider the reference Brownian motions to be in stationarity. We will make use of properties of the KL divergence defined relative to these measures; see the Appendix of Léonard (2013) for a justification of why these properties still hold when the reference measures are unbounded.

We first provide a result detailing the relationships between the dynamic and static tree-structured SB problems.

**Proposition B.1** (Compare to Proposition 2.3 in Léonard (2013))**.** *The tree-structured Schrödinger Bridge problems* (TreeSB$_{\text{dyn}}$)*,* (TreeSB$_{\text{stat}}$)*, and* (TreeSB$_{\text{stat}}^{\mathcal{S}}$) *admit at most one solution* $\mathbb{P}^{SB} \in \mathcal{P}(\mathcal{C}_{\mathcal{T}})$*,* $\Pi_{\mathcal{V}}^{SB} \in \mathcal{P}((\mathbb{R}^d)^{|\mathcal{V}|})$*, and* $\Pi_{\mathcal{S}}^{SB} \in \mathcal{P}((\mathbb{R}^d)^{|\mathcal{S}|})$ *respectively.*

*If* $\mathbb{P}^{SB}$ *solves* (TreeSB$_{\text{dyn}}$)*, then* $\mathbb{P}_{\mathcal{V}}^{SB}$ *solves* (TreeSB$_{\text{stat}}$)*. Conversely, if* $\Pi_{\mathcal{V}}^{SB}$ *solves* (TreeSB$_{\text{stat}}$)*, then* (TreeSB$_{\text{dyn}}$) *is solved by mixing Brownian bridges along each edge as* $\mathbb{P}^{SB} = \Pi_{\mathcal{V}}^{SB} \mathbb{Q}_{\cdot|\mathcal{V}}$*.*

*Moreover, if* $\mathbb{P}^{SB}$ *solves* (TreeSB$_{\text{dyn}}$)*, then* $\mathbb{P}_{\mathcal{S}}^{SB}$ *solves* (TreeSB$_{\text{stat}}^{\mathcal{S}}$)*. Conversely, if* $\Pi_{\mathcal{S}}^{SB}$ *solves* (TreeSB$_{\text{stat}}^{\mathcal{S}}$)*, then* (TreeSB$_{\text{dyn}}$) *is solved by the corresponding mixture of* $\mathbb{Q}$*-bridges conditioned on values at* $\mathcal{S}$*,* $\mathbb{P}^{SB} = \Pi_{\mathcal{S}}^{SB} \mathbb{Q}_{\cdot|\mathcal{S}}$*.*

*Proof.* The first statement follows from the strict convexity of the (TreeSB$_{\text{stat}}$), (TreeSB$_{\text{dyn}}$), and (TreeSB$_{\text{stat}}^{\mathcal{S}}$) problems.

The second follows by using the chain rule for KL divergence, conditioning on the values at the vertices $\mathcal{V}$. We obtain

$$\mathrm{KL}(\mathbb{P}\|\mathbb{Q}) = \mathrm{KL}(\mathbb{P}_{\mathcal{V}}\|\mathbb{Q}_{\mathcal{V}}) + \int_{(\mathbb{R}^d)^{|\mathcal{V}|}} \mathrm{KL}(\mathbb{P}(\cdot|X_{\mathcal{V}})\|\mathbb{Q}(\cdot|X_{\mathcal{V}}))\mathrm{d}\mathbb{P}_{\mathcal{V}}(X_{\mathcal{V}}). \qquad (22)$$

We see that $\mathrm{KL}(\mathbb{P}\|\mathbb{Q}) \geq \mathrm{KL}(\mathbb{P}_{\mathcal{V}}\|\mathbb{Q}_{\mathcal{V}})$, with equality if and only if $\mathbb{P}(\cdot|X_{\mathcal{V}}) = \mathbb{Q}(\cdot|X_{\mathcal{V}})$ for $\mathbb{P}_{\mathcal{V}}$-a.e. $X_{\mathcal{V}}$ (assuming $KL(\mathbb{P}\|\mathbb{Q}) < \infty$). Therefore, $\mathbb{P}$ solves the dynamic problem if and only if it decomposes as a mixture over bridges $\mathbb{Q}_{\cdot|\mathcal{V}}$ according to the coupling $\mathbb{P}_{\mathcal{V}}$ solving the static problem, i.e. $\mathbb{P}^{SB} = \Pi_{\mathcal{V}}^{SB} \mathbb{Q}_{\cdot|\mathcal{V}}$ (if this were not true, then $\mathbb{P}_{\mathcal{V}}^{SB}\mathbb{Q}_{\cdot|\mathcal{V}}$ would be a valid solution with lower KL divergence relative to $\mathbb{Q}$, contradicting optimality of $\mathbb{P}^{SB}$). Note that such bridges just consist of Brownian bridges along the individual edges.

The third part follows similarly to the second. Consider the KL decomposition but instead conditioning only on the values on the observed vertices $\mathcal{S}$,

$$\mathrm{KL}(\mathbb{P}\|\mathbb{Q}) = \mathrm{KL}(\mathbb{P}_{\mathcal{S}}\|\mathbb{Q}_{\mathcal{S}}) + \int_{(\mathbb{R}^d)^{|\mathcal{S}|}} \mathrm{KL}(\mathbb{P}(\cdot|X_{\mathcal{S}})\|\mathbb{Q}(\cdot|X_{\mathcal{S}}))\mathrm{d}\mathbb{P}_S(X_{\mathcal{S}}). \qquad (23)$$

Now we have $\mathrm{KL}(\mathbb{P}\|\mathbb{Q}) \geq \mathrm{KL}(\mathbb{P}_{\mathcal{S}}\|\mathbb{Q}_{\mathcal{S}})$, with equality if and only if $\mathbb{P}(\cdot|X_{\mathcal{S}}) = \mathbb{Q}(\cdot|X_{\mathcal{S}})$ for $\mathbb{P}_{\mathcal{S}}$-a.e. $X_{\mathcal{S}}$ (assuming $\mathrm{KL}(\mathbb{P}\|\mathbb{Q}) < \infty$). So in particular, we have that $\mathbb{P}^{SB}$ must be a mixture of $\mathbb{Q}$-bridges according to its own coupling over $\mathcal{S}$, i.e. $\mathbb{P}^{SB} = \mathbb{P}_{\mathcal{S}}^{SB}\mathbb{Q}_{\cdot|\mathcal{S}}$. $\qquad \square$

In the following, we will utilise this equivalence between the $(\text{TreeSB}_{\text{dyn}})$ and $(\text{TreeSB}_{\text{stat}}^{\mathcal{S}})$ problems. We now present an auxiliary result giving a criterion for the tree-structured SB problems to have a solution, in the vein of Lemma 2.4 in Léonard (2013). This is a technical result required to deal with the fact that we are considering an unbounded reference measure $\mathbb{Q}$.

**Lemma B.2** (Compare to Lemma 2.4 in Léonard (2013)). *Let $B : \mathbb{R}^d \to [0, \infty)$ be a measurable function such that*

$$z_B := \int_{(\mathbb{R}^d)^{|\mathcal{S}|}} \exp\Big( - \sum_{i \in \mathcal{S}} B(x_i) \Big) \mathbb{Q}_{\mathcal{S}}(\mathrm{d}x_{\mathcal{S}}) < \infty, \tag{24}$$

*and for each $i \in \mathcal{S}$ take a $\mu_i \in \mathcal{P}(\mathcal{R}^d)$ such that $\int B \mathrm{d}\mu_i < \infty$.*

*Note that $\inf(\text{TreeSB}_{\text{stat}}^{\mathcal{S}}) = \inf(\text{TreeSB}_{\text{dyn}}) \in (-\infty, \infty]$ (from the previous result). The static and dynamic tree-structured SB problems $(\text{TreeSB}_{\text{stat}}^{\mathcal{S}})$ and $(\text{TreeSB}_{\text{dyn}})$ for the $\mu_i$ have a (unique) solution if and only if $\inf(\text{TreeSB}_{\text{stat}}^{\mathcal{S}}) = \inf(\text{TreeSB}_{\text{dyn}}) < \infty$ (that is, if and only if the marginals $\mu_i$ are such that there exists some $\Pi^0 \in \mathcal{P}((\mathbb{R}^d)^{|\mathcal{S}|})$ satisfying $\Pi_i^0 = \mu_i$ for each $i \in \mathcal{S}$, and $\mathrm{KL}(\Pi^0 \| \mathbb{Q}_{\mathcal{S}}) < \infty$).*

*Proof.* In light of the equivalence in Proposition B.1, we can consider just the static problem $(\text{TreeSB}_{\text{stat}}^{\mathcal{S}})$. Since the marginals $\mu_i$ are tight on $\mathbb{R}^d$, it easily follows that the closed constraint set $\Gamma(\{\mu_i\}_{i \in \mathcal{S}}) := \{\Pi \in \mathcal{P}((\mathbb{R}^d)^{|\mathcal{S}|} : \Pi_i = \mu_i \quad \forall i \in \mathcal{S}\}$ is uniformly tight and thus compact in $\mathcal{P}((\mathbb{R}^d)^{|\mathcal{S}|})$. From the characterisation of KL divergence with respect to the unbounded measure $\mathbb{Q}_{\mathcal{S}}$ (see Léonard (2013), Appendix A), we have $\mathrm{KL}(\Pi \| \mathbb{Q}_{\mathcal{S}}) = \mathrm{KL}(\Pi \| \mathbb{Q}_{\mathcal{S}}^B) - \int_{(\mathbb{R}^d)^{|\mathcal{S}|}} \sum_{i \in \mathcal{S}} B(x_i) \mathrm{d}\Pi(x_{\mathcal{S}}) - z_B$, where $\mathbb{Q}_{\mathcal{S}}^B$ is the normalised measure $\mathbb{Q}_{\mathcal{S}}^B = \frac{1}{z_B} \exp(- \bigoplus_{\mathcal{S}} B) \mathbb{Q}_{\mathcal{S}}$. For $\Pi \in \Gamma(\{\mu_i\}_{i \in \mathcal{S}})$, we have

$$\mathrm{KL}(\Pi \| \mathbb{Q}_{\mathcal{S}}) = \mathrm{KL}(\Pi \| \mathbb{Q}_{\mathcal{S}}^B) - \sum_{i \in \mathcal{S}} \int_{\mathbb{R}^d} B \mathrm{d}\mu_i - z_B.$$

The lower-semicontinuity of $\Pi \mapsto \mathrm{KL}(\Pi \| \mathbb{Q}_{\mathcal{S}}^B)$, and the assumption $\int B \mathrm{d}\mu_i < \infty$ for each $i \in \mathcal{S}$, together imply that $\mathrm{KL}(\Pi \| \mathbb{Q}_{\mathcal{S}})$ is lower bounded and lower semi-continuous on the compact set $\Gamma(\{\mu_i\}_{i \in \mathcal{S}})$. Thus the static problem $(\text{TreeSB}_{\text{stat}}^{\mathcal{S}})$ admits a solution if and only if $\inf(\text{TreeSB}_{\text{stat}}^{\mathcal{S}}) < \infty$. $\qquad \square$

In light of Lemma B.2, we can state the following result which provides conditions for the existence of the $(\text{TreeSB}_{\text{dyn}})$ solution. Recall we will consider an unnormalised Brownian reference measure, which satisfies $\mathbb{Q}_i = \text{Leb}$ for each $i \in \mathcal{S}$, so the following result applies with $m = \text{Leb}$.

**Proposition B.3** (Compare to Proposition 2.5 in Léonard (2013)). *Suppose that $\mathbb{Q}_i = m$ for each $i \in \mathcal{S}$, for a positive measure $m$. We have the following results.*

(a) *For $(\text{TreeSB}_{\text{stat}}^{\mathcal{S}})$ and $(\text{TreeSB}_{\text{dyn}})$ to have a solution, it is necessary to have $\mathrm{KL}(\mu_i \| m) < \infty$ for each $i \in \mathcal{S}$.*

(b) *For sufficient conditions: Suppose there exists measurable functions $A, B : \mathbb{R}^d \to \mathbb{R}^{\geq 0}$ satisfying*

  (i) $\mathbb{Q}_{\mathcal{S}}(\mathrm{d}x_{\mathcal{S}}) \geq \exp\big( - \sum_{i \in \mathcal{S}} A(x_i) \big) \mathrm{d}m^{\otimes s}(x_{\mathcal{S}})$,

  (ii) $\int_{(\mathbb{R}^d)^{|\mathcal{S}|}} \exp\big( - \sum_{i \in \mathcal{S}} B(x_i) \big) \mathbb{Q}_{\mathcal{S}}(\mathrm{d}x_{\mathcal{S}}) < \infty$,

  (iii) $\int_{\mathbb{R}^d} (A + B) \mathrm{d}\mu_i < \infty$ for each $i \in \mathcal{S}$.

  (iv) $\mathrm{KL}(\mu_i \| m) < \infty$ for each $i \in \mathcal{S}$.

  *Then there exists a unique solution to the SB problems $(\text{TreeSB}_{\text{stat}}^{\mathcal{S}})$ and $(\text{TreeSB}_{\text{dyn}})$.*

*Proof.* The first statement (a) follows by applying the result in Lemma B.2 and using the fact that $\mathrm{KL}(\mu_i \| \text{Leb}) \leq \mathrm{KL}(\Pi^0 \| \mathbb{Q}_{\mathcal{S}}) < \infty$.

For the second, we take the independent coupling $\Pi^0_{\mathcal{S}} = \bigotimes_{i \in \mathcal{S}} \mu_i$, and observe that (under the above assumptions) it satisfies the conditions in Lemma B.2. Clearly, it has the correct marginals $\Pi^0_i = \mu_i$ by construction, so it remains only to check that $\mathrm{KL}(\Pi^0_{\mathcal{S}} \| \mathbb{Q}_{\mathcal{S}}) < \infty$. By expanding the KL divergence and using the inequalities in (i) and (iii), one sees that the inequalities in (iv) are sufficient to ensure $\mathrm{KL}(\Pi^0_{\mathcal{S}} \| \mathbb{Q}_{\mathcal{S}}) < \infty$. $\qquad\square$

## B.2 Characterisation of TreeSB solution

We now show the characterisation of the solution to Equation (TreeSB$_{\mathrm{dyn}}$) stated in Theorem 3.1, upon which the IMF procedure depends. We first recall that the solution $\mathbb{P}^{SB}$ is a mixture of bridges of $\mathbb{Q}$, conditional on the values at vertices in $\mathcal{S}$.

**Proposition B.4** (**Reciprocal process**). *The solution $\mathbb{P}^{SB}$ to (TreeSB$_{\mathrm{dyn}}$) (if it exists) is in the reciprocal class $\mathcal{R}_{\mathcal{S}}(\mathbb{Q})$, i.e. $\mathbb{P}^{SB} = \mathbb{P}^{SB}_{\mathcal{S}} \mathbb{Q}_{\cdot | \mathcal{S}}$.*

*Proof.* This is a consequence of the third part of Proposition B.1. $\qquad\square$

We now show that for a Markov reference process $\mathbb{Q}$, the (TreeSB$_{\mathrm{dyn}}$) solution is also Markov. Following Léonard (2013), to present the results we will use the following characterisation of Marko-vianity for a path measure $\mathbb{P}$ on the tree structure. Consider an edge $e \in \mathcal{E}$ and a time $t_e \in [0, T^e]$. Consider the time $t_e$ as splitting the continuous tree into two distinct sections (note that such a splitting may not be unique, as the chosen point may correspond to a vertex; in such cases consider any such split into two distinct parts). Denote the restrictions of a process $X$ to these two parts as $X_{\leq}$ and $X_{>}$. Then we say $\mathbb{P}$ is Markov to mean that $\mathbb{P}(X_{\leq} \in \cdot, X_{\geq} \in \cdot \cdot | X^e_t) = \mathbb{P}(X_{\leq} \in \cdot | X^e_t) \mathbb{P}(X_{\geq} \in \cdot \cdot | X^e_t)$ for any such split (together with the technical assumption that some time-marginal of $\mathbb{P}$ is $\sigma$-finite; see discussion in Léonard (2013)). Note that the Brownian reference process considered in the main paper is Markov.

**Proposition B.5** (**Markov process**). *For Markov reference measure $\mathbb{Q}$, the solution $\mathbb{P}^{SB}$ to (TreeSB$_{\mathrm{dyn}}$) (if it exists) is Markov.*

*Proof.* The proof follows that of Proposition 2.10 in Léonard (2013). We outline the following changes to the notation for our setting, then the argument follows the same way.

Consider an edge $e \in \mathcal{E}$ and a time $t_e \in [0, T^e]$, along with a corresponding split of the tree at time $t_e$ as described above. We define the following notation: Let $\mathcal{C}^{\leq}_{\mathcal{T}} = \{\omega^{\leq} : \omega \in \mathcal{C}_{\mathcal{T}}\}$ and $\mathcal{C}^{\geq}_{\mathcal{T}} = \{\omega^{\geq} : \omega \in \mathcal{C}_{\mathcal{T}}\}$ be the spaces of continuous paths on the two sections of the tree respectively. For a path measure $\mathbb{P} \in \mathcal{P}(\mathcal{C}_{\mathcal{T}})$, let $\mathbb{P}^{t_e, z} = \mathbb{P}(\cdot | X^e_{t_e} = z) \in \mathcal{P}(\mathcal{C}_{\mathcal{T}})$ be the measure conditioned on the process $X$ taking value $x$ at the time $t_e$, and moreover define its restrictions to the two sections as $\mathbb{P}^{t_e, z}_{\leq t_e}$ and $\mathbb{P}^{t_e, z}_{\leq t_e}$ respectively.

Similarly to Léonard (2013) we now make the following claim, from which the result follows.

**Claim B.6.** *Fix a time on the tree $t_e$ as described above. Fix a $z \in \mathbb{R}^d$, a measure $\mu \in \mathcal{P}(\mathbb{R}^d)$, and path measures on the 'before' and 'after' sections $\tilde{\mathbb{P}}^{t_e, z}_{\leq} \in \mathcal{P}(\mathcal{C}^{\leq}_{\mathcal{T}} \cap \{X_{t_e} = z\})$ and $\tilde{\mathbb{P}}^{t_e, z}_{\geq} \in \mathcal{P}(\mathcal{C}^{\geq}_{\mathcal{T}} \cap \{X_{t_e} = z\})$. Consider minimising $\mathrm{KL}(\cdot \| \mathbb{Q})$ over path measures $\mathbb{P} \in \mathcal{P}(\mathcal{C}_{\mathcal{T}})$ constrained to satisfy $\mathbb{P}_{t_e} = \mu$, $\mathbb{P}^{t_e, z}_{\leq} = \tilde{\mathbb{P}}^{t_e, z}_{\leq}$, and $\mathbb{P}^{t_e, z}_{\geq} = \tilde{\mathbb{P}}^{t_e, z}_{\geq}$. Then the objective $\mathrm{KL}(\cdot \| \mathbb{Q})$ attains its unique minimum at $\mathbb{P}^*(\cdot) = \int_{\mathbb{R}^d} \tilde{\mathbb{P}}^{t_e, z}_{\leq} \otimes \tilde{\mathbb{P}}^{t_e, z}_{\geq} \mu(\mathrm{d}z)$.*

Given the claim, the result follows according to the following argument: Suppose for a contradiction that the SB solution, here denoted $\tilde{\mathbb{P}}$, was not Markov. Then, there exists some time $t_e$ and a correspond split of the tree such that $\tilde{\mathbb{P}}(\cdot | X^e_{t_e}) \neq \tilde{\mathbb{P}}_{\leq}(\cdot | X^e_{t_e}) \otimes \tilde{\mathbb{P}}_{\geq}(\cdot | X^e_{t_e})$. Applying the above claim with $\mu = \tilde{\mathbb{P}}_{t_e}$, we see that $\mathbb{P}^*$ and $\tilde{\mathbb{P}}$ have the same marginals at all time-points on the tree, but $\mathbb{P}^*$ attains a strictly lower KL divergence $\mathrm{KL}(\mathbb{P}^* \| \mathbb{Q}) < \mathrm{KL}(\tilde{\mathbb{P}} \| \mathbb{Q})$, contradicting the optimality of $\tilde{\mathbb{P}}$.

The proof of the claim uses Jensen's inequality and proceeds exactly as the proof of Claim 2.11 in Léonard (2013), with the appropriate notation changes. $\qquad\square$

We now provide the characterisation of the TreeSB solution in Theorem 3.1. While the previous results have been for a general reference measure $\mathbb{Q}$, we present the following results for $\mathbb{Q}$ associated to running Brownian motions $(\sigma B_t^e)_{t \in [0, T^e]}$ along each edge, as considered in the main paper.

**Theorem 3.1** (**TreeSB characterisation**). *Under mild assumptions (in the Brownian case, namely that $\int \|x\|^2 \mathrm{d}\mu_i(x) < \infty$ and $\mathrm{H}(\mu_i) < \infty$ for each $i \in \mathcal{S}$), there exists a unique solution to the dynamic TreeSB problem* (TreeSB$_{\mathrm{dyn}}$). *The solution is the unique process $\mathbb{P}$ that is both Markov and in $\mathcal{R}_\mathcal{S}(\mathbb{Q})$ with correct marginals $\mathbb{P}_i = \mu_i$ for $i \in \mathcal{S}$.*

*Proof.* As we are considering a Brownian reference process, the assumptions in question are that $\int \|x\|^2 \mathrm{d}\mu_i(x) < \infty$, and $\mathrm{H}(\mu_i) < \infty$ for each $i \in \mathcal{S}$. One can then verify (as in De Bortoli et al. (2024), Lemma D.2) that the criteria in Proposition B.3 hold by taking functions $A$ and $B$ to be quadratic, from which uniqueness and existence of the solution follow. From Proposition B.4 we have that the solution is in $\mathcal{R}_\mathcal{S}(\mathbb{Q})$, and from Proposition B.5 we have that it is Markov.

We now need to show that if a measure $\mathbb{P}^0$ is Markov and in $\mathcal{R}_\mathcal{S}(\mathbb{Q})$, and has the correct marginals $\mathbb{P}_i^0 = \mu_i$ for $i \in \mathcal{S}$, then it is the TreeSB solution. Note first that as $\mathbb{P}^0$ is Markov and reciprocal, its restriction to each edge $e = (u, v)$ is also Markov and reciprocal along that edge. Thus, by Theorem 2.14 in Léonard et al. (2014) (noting that the required criterion holds for the Brownian reference measure) we have that $\frac{\mathrm{d}\mathbb{P}_e^0}{\mathrm{d}\mathbb{Q}_e} = f_u(X_u)f_v(X_v)$, $\mathbb{Q}_e$-a.e. for some non-negative measurable functions $f_u, f_v$. Recall that the path measures are a composition of the path measures along each edge according to the tree structure, so this means that $\frac{\mathrm{d}\mathbb{P}^0}{\mathrm{d}\mathbb{Q}} = \prod_{i \in \mathcal{V}} f_i(X_i)$, $\mathbb{Q}$-a.e. for some non-negative measurable functions $f_i$ (via relabelling of the functions). Note too that $\mathbb{P}^0$ is in $\mathcal{R}_\mathcal{S}(\mathbb{Q})$, so we can also express the Radon-Nikodym derivative as $\frac{\mathrm{d}\mathbb{P}^0}{\mathrm{d}\mathbb{Q}} = h(\{X_i\}_{i \in \mathcal{S}})$ for some non-negative measurable function $h$. Equating the two expressions, we see that we in fact must have a decomposition only over vertices in $\mathcal{S}$, $\frac{\mathrm{d}\mathbb{P}^0}{\mathrm{d}\mathbb{Q}} = \frac{\mathrm{d}\Pi^0}{\mathrm{d}\mathbb{Q}_\mathcal{V}} = \prod_{i \in \mathcal{S}} f_i(X_i)$, $\mathbb{Q}$-a.e. for some non-negative measurable functions $f_i$ (where $\Pi^0$ denotes the static coupling of $\mathbb{P}^0$ over the vertices $\mathcal{V}$).

The remainder follows the standard argument characterising the SB solution using the decomposition according to potentials (see e.g. Nutz (2021)). Consider static couplings in the constraint set $\Pi \in \Gamma(\{\mu_i\}_{i \in \mathcal{S}}) := \{\Pi \in \mathcal{P}((\mathbb{R}^d)^{|\mathcal{V}|} : \Pi_i = \mu_i \ \forall i \in \mathcal{S}\}$ such that $\mathrm{KL}(\Pi|\mathbb{Q}_\mathcal{V}) < \infty$. By the above, we have that $\mathbb{E}_\Pi[\log(\frac{\mathrm{d}\Pi^0}{\mathrm{d}\mathbb{Q}_\mathcal{V}})] = \sum_{i \in \mathcal{S}} \int \log f_i \mathrm{d}\mu_i$, which is in particular independent of the choice of $\Pi$ (for a precise statement taking care regarding the integrability of the potentials, follow the argument of Proposition 2.17 in Nutz (2021)). Therefore, for any such $\Pi$ we have

$$\mathrm{KL}(\Pi\|\mathbb{Q}_\mathcal{V}) \geq \mathrm{KL}(\Pi\|\mathbb{Q}_\mathcal{V}) - \mathrm{KL}(\Pi\|\Pi^0) \tag{25}$$

$$= \mathbb{E}_\Pi[\log(\tfrac{\mathrm{d}\Pi^0}{\mathrm{d}\mathbb{Q}_\mathcal{V}})] \tag{26}$$

$$= \mathbb{E}_{\Pi_0}[\log(\tfrac{\mathrm{d}\Pi^0}{\mathrm{d}\mathbb{Q}_\mathcal{V}})] \tag{27}$$

$$= \mathrm{KL}(\Pi^0\|\mathbb{Q}_\mathcal{V}). \tag{28}$$

and we thus see that $\Pi^0$ is the minimiser of the (TreeSB$_{\mathrm{stat}}$) problem. $\qquad\square$

## B.3 Properties of the tree-structured projections

We now move on to proving properties of the TreeIMF procedure. We follow the presentation of Shi et al. (2023). We begin by proving the properties of the reciprocal and Markovian projections defined in Definitions 3.3 and 3.4. For a full set of required assumptions, see the assumptions A.1, A.2 and A.3 in Shi et al. (2023) Appendix C.2, which are standard in the literature and we assume to hold along each edge.

The following result follows similarly to the standard IMF case.

**Proposition 3.3.** *For $\mathbb{P} \in \mathcal{P}(\mathcal{C}_\mathcal{T})$, the reciprocal projection $\Pi^* = \mathrm{proj}_{\mathcal{R}_\mathcal{S}(\mathbb{Q})}(\mathbb{P})$ solves the minimisation problem $\Pi^* = \arg\min_{\Pi \in \mathcal{R}_\mathcal{S}(\mathbb{Q})} \mathrm{KL}(\mathbb{P}\|\Pi)$.*

*Proof.* This follows from the KL decomposition used in Proposition B.1, conditioning on the values on the observed vertices $\mathcal{S}$:

$$\mathrm{KL}(\mathbb{P}\|\Pi) = \mathrm{KL}(\mathbb{P}_\mathcal{S}\|\Pi_\mathcal{S}) + \int_{(\mathbb{R}^d)^{|\mathcal{S}|}} \mathrm{KL}(\mathbb{P}(\cdot|X_\mathcal{S})\|\Pi(\cdot|X_\mathcal{S}))\mathrm{d}\mathbb{P}_\mathcal{S}(X_\mathcal{S}). \tag{29}$$

Given we are optimising $\Pi$ over the reciprocal class $\mathcal{R}_\mathcal{S}(\mathbb{Q})$, we have that the bridges $\Pi(\cdot|X_\mathcal{S}) = \mathbb{Q}(\cdot|X_\mathcal{S})$ are fixed. Thus, the minimiser is achieved by taking $\Pi_\mathcal{S} = \mathbb{P}_\mathcal{S}$, that is $\Pi = \mathrm{proj}_{\mathcal{R}_\mathcal{S}(\mathbb{Q})}(\mathbb{P})$.
□

To prove subsequent results, we require the following decomposition of the KL divergence according to the tree structure. Note that we consider Markov and reciprocal processes on the tree, both of which factorise according to the tree structure so the following decomposition can be applied.

**Lemma 3.2** (**KL decomposition along tree**). *Take path measures $\mathbb{P}, \tilde{\mathbb{P}}$ that share a marginal at the root, $\mathbb{P}_r = \tilde{\mathbb{P}}_r$, and factorise along the tree. Under mild assumptions, we have the KL decomposition*

$$\mathrm{KL}(\mathbb{P}\|\tilde{\mathbb{P}}) = \sum_{(u,v)\in\mathcal{E}_r} \mathbb{E}_{X_u\sim\mathbb{P}_u}\Big[\mathrm{KL}(\mathbb{P}^{(u,v)}(\cdot|X_u)\|\tilde{\mathbb{P}}^{(u,v)}(\cdot|X_u))\Big]. \tag{12}$$

*Proof.* This is a consequence of the iterative application of the chain rule for KL divergence applied according to the tree structure, and the conditional independence caused by the tree structure. By first applying the chain rule for the KL divergence conditional on the value at the root vertex $r$, we have

$$\mathrm{KL}(\mathbb{P}\|\tilde{\mathbb{P}}) = \mathrm{KL}(\mathbb{P}_r\|\tilde{\mathbb{P}}_r) + \mathbb{E}_{X_r\sim\mathbb{P}_r}\big[\mathrm{KL}\big(\mathbb{P}(\cdot|X_r)\|\tilde{\mathbb{P}}(\cdot|X_r)\big)\big] \tag{30}$$

$$= \mathbb{E}_{X_r\sim\mathbb{P}_r}\big[\mathrm{KL}\big(\mathbb{P}(\cdot|X_r)\|\tilde{\mathbb{P}}(\cdot|X_r)\big)\big]. \tag{31}$$

Recall that the edge set $\mathcal{E}$ is depth-wise ordered. We can again apply the KL chain rule to the term inside the expectation, now conditioned on the process along the first edge $e_1$.

$$\mathrm{KL}\big(\mathbb{P}(\cdot|X_r)\|\tilde{\mathbb{P}}(\cdot|X_r)\big) = \mathrm{KL}\big(\mathbb{P}^{e_1}(\cdot|X_r)\|\tilde{\mathbb{P}}^{e_1}(\cdot|X_r)\big)$$
$$+ \mathbb{E}_{X^{e_1}\sim\mathbb{P}^{e_1}(\cdot|X_r)}\big[\mathrm{KL}\big(\mathbb{P}(\cdot|X^{e_1},X_r)\|\tilde{\mathbb{P}}(\cdot|X^{e_1},X_r)\big)\big]. \tag{32}$$

We can iteratively apply similar decompositions as we traverse the edges according to the ordered edge set.

$$\mathrm{KL}\big(\mathbb{P}(\cdot|X^{e_k},...,X^{e_1},X_r)\|\tilde{\mathbb{P}}(\cdot|X^{e_k},...,X^{e_1},X_r)\big)$$
$$= \mathrm{KL}\big(\mathbb{P}^{e_{k+1}}(\cdot|X^{e_k},...,X^{e_1},X_r)\|\tilde{\mathbb{P}}^{e_{k+1}}(\cdot|X^{e_k},...,X^{e_1},X_r)\big)$$
$$+ \mathbb{E}_{X^{e_{k+1}}\sim\mathbb{P}^{e_{k+1}}(\cdot|X^{e_k},...,X^{e_1},X_r)}\big[\mathrm{KL}\big(\mathbb{P}(\cdot|X^{e_{k+1}},...,X^{e_1},X_r)\|\tilde{\mathbb{P}}(\cdot|X^{e^{k+1}},...,X^{e_1},X_r)\big)\big] \tag{33}$$

$$= \mathrm{KL}\big(\mathbb{P}^{e_{k+1}}(\cdot|X_{s(e_{k+1})})\|\tilde{\mathbb{P}}^{e_{k+1}}(\cdot|X_{s(e_{k+1})})\big)$$
$$+ \mathbb{E}_{X^{e_{k+1}}\sim\mathbb{P}^{e_{k+1}}(\cdot|X^{e_k},...,X^{e_1},X_r)}\big[\mathrm{KL}\big(\mathbb{P}(\cdot|X^{e_{k+1}},...,X^{e_1},X_r)\|\tilde{\mathbb{P}}(\cdot|X^{e_{k+1}},...,X^{e_1},X_r)\big)\big], \tag{34}$$

where in the second line we use the factorisation property along the tree structure (here, $s(e)$ denotes the starting vertex of an edge $e$). Applying such decompositions for each edge in the ordered edge set $\mathcal{E}$, one obtains

$$\mathrm{KL}(\mathbb{P}\|\tilde{\mathbb{P}}) = \sum_{k=1}^{|\mathcal{E}|} \mathbb{E}_{X_{e_{k-1}}\sim\mathbb{P}^{e_{k-1}}(\cdot|X^{e_{k-2}},...,X^{e_1},X_r)}\Big[\mathrm{KL}\big(\mathbb{P}^{e_k}(\cdot|X_{s(e_k)})\|\tilde{\mathbb{P}}^{e_k}(\cdot|X_{s(e_k)})\big)\Big] \tag{35}$$
$$\vdots$$
$$X_{e_1}\sim\mathbb{P}^{e_1}(\cdot|X_r)$$
$$X_r\sim\mathbb{P}_r$$
$$= \sum_{(u,v)\in\mathcal{E}_r} \mathbb{E}_{X_u\sim\mathbb{P}_u}\Big[\mathrm{KL}(\mathbb{P}^{(u,v)}(\cdot|X_u)\|\tilde{\mathbb{P}}^{(u,v)}(\cdot|X_u))\Big]. \tag{36}$$

as required.
□

We now consider the Markov projection defined in Definition 3.4. Note that the restriction of the TreeSB solution to each edge is itself an SB (because it is Markov and reciprocal), and thus can be associated with an SDE (Dai Pra, 1991). In the definition of the Markov class in Definition 3.1 we define the Markov class $\mathcal{M}_{\mathcal{T}}$ via considering SDEs with locally Lipschitz drifts. The restriction to locally Lipschitz drifts is a technical requirement for applying the entropic version of Girsanov's theorem; this is standard in the literature and does not affect our methodology. Following Shi et al. (2023), we now provide a result showing that the Markov projection also solves a minimisation problem.

**Proposition 3.4.** *Under mild assumptions, the Markovian projection $\mathbb{M}^* = \mathrm{proj}_{\mathcal{M}}(\Pi)$ solves the minimisation problem $\mathbb{M}^* = \arg\min_{\mathbb{M} \in \mathcal{M}_{\mathcal{T}}} \mathrm{KL}(\Pi\|\mathbb{M})$.*

*Proof.* Applying the KL decomposition in Lemma 3.2, we have

$$\mathrm{KL}(\Pi\|\mathbb{M}) = \sum_{e=(u,v)\in\mathcal{E}} \mathbb{E}_{X_u \sim \Pi_u}\Big[\mathrm{KL}(\Pi^e(\cdot|X_u)\|\mathbb{M}^e(\cdot|X_u))\Big]. \tag{37}$$

We now analyse the individual KL expressions $\mathrm{KL}(\Pi^e(\cdot|X_u)\|\mathbb{M}^e(\cdot|X_u))$ along each edge $e = (u,v)$, using the proof techniques of Proposition 2 in Shi et al. (2023).

In particular, applying the argument in the proof of Proposition 2 in Shi et al. (2023), one sees that each conditional process $\Pi^e_{\cdot|0}$ is Markov and can be associated with $(X_t^e)_{t\in[0,T^e]}$ given by

$$X_t^e = \sigma^2 \int_0^t \mathbb{E}_{\Pi^e_{T^e|s,0}}[\nabla \log \mathbb{Q}^e_{T^e|s}(X^e_{T^e}|X_s^e)|X_s^e, X_0^e]\mathrm{d}s + \sigma \int_0^t \mathrm{d}B_s^e. \tag{38}$$

Therefore, letting the restriction of the Markov process to edge $e$ (denoted above as $\mathbb{M}^e$) be associated with a process $\mathrm{d}Y_t^e = v_e(t, Y_t^e)\mathrm{d}t + \sigma\mathrm{d}B_t^e$ such that $\mathrm{KL}(\Pi^e(\cdot|X_u)\|\mathbb{M}^e(\cdot|X_u)) < \infty$, with $v_e$ locally Lipschitz, then (using e.g. Léonard (2012), Theorem 2.3) one obtains

$$\mathrm{KL}(\Pi^e(\cdot|X_u)\|\mathbb{M}^e(\cdot|X_u)) = \tfrac{1}{2\sigma^2} \int_0^{T^e} \mathbb{E}_{\Pi^e_{t|0}}\big[\|\sigma^2\mathbb{E}_{\Pi^e_{T^e|t,0}}[\nabla \log \mathbb{Q}_{T^e|t}(X^e_{T^e}|X_t^e)|X_t^e, X_0^e]$$
$$-v_e(t, X_t^e)\|^2\big]\mathrm{d}t. \tag{39}$$

Thus, substituting back into (37) we have

$$\mathrm{KL}(\Pi\|\mathbb{M}) = \sum_{e=(u,v)\in\mathcal{E}} \mathbb{E}_{X_u \sim \Pi_u}\Big[\mathrm{KL}(\Pi^e(\cdot|X_u)\|\mathbb{M}^e(\cdot|X_u))\Big] \tag{40}$$

$$= \sum_{e\in\mathcal{E}} \mathbb{E}_{X_u \sim \Pi_u}\Big[\tfrac{1}{2\sigma^2} \int_0^{T_e} \mathbb{E}_{\Pi^e_{t|0}}\big[\|\sigma^2\mathbb{E}_{\Pi^e_{T|0,t}}[\nabla \log \mathbb{Q}^e_{T|t}(X^e_T|X_t^e)|X_t^e, X_0^e] - v_e(X_t^e, t)\|^2\big]\mathrm{d}t\Big] \tag{41}$$

$$= \sum_{e\in\mathcal{E}} \tfrac{1}{2\sigma^2} \int_0^{T_e} \mathbb{E}_{X_0, X_t \sim \Pi^e_{0,t}}\big[\|\sigma^2\mathbb{E}_{\Pi^e_{T|0,t}}[\nabla \log \mathbb{Q}^e_{T|t}(X^e_T|X_t^e)|X_t^e, X_0^e] - v_e(X_t^e, t)\|^2\big]\mathrm{d}t. \tag{42}$$

This expression is minimised by taking $v_e^*(t, x) = \sigma^2\mathbb{E}_{\Pi^e_{T|t}}\big[\nabla \log \mathbb{Q}^e_{T|t}(X^e_T|X_t^e)|X_t^e = x\big]$ along each edge $e \in \mathcal{E}$. This corresponds to performing a Markovian projection along each edge $e$ according to the coupling $\Pi^e$, which is exactly the definition of the tree-based Markovian projection in Definition 3.4.

We also note that, as an instance of bridge matching, along each edge the process $\Pi_t^e$ and its corresponding Markovian projection $\mathbb{M}_t^{e,*}$ satisfy the same Fokker-Planck equation (Peluchetti (2022), Theorem 2). Thus by the uniqueness of the solutions of the Fokker-Planck equations under A.1 and A.3 in Shi et al. (2023) (see e.g. Bogachev et al. (2021)), they share the same marginals $\mathbb{M}_t^{e,*} = \Pi_t^e$. $\qquad\square$

## B.4 TreeIMF convergence

We follow the presentation of convergence in Shi et al. (2023), but with the appropriate modifications to the proofs for the tree-based setting.

**Lemma B.7** (**Pythagorean property**, compare to Shi et al. (2023), Lemma 6). *Take a Markovian process* $\mathbb{M} \in \mathcal{M}_{\mathcal{T}}$ *and a reciprocal process* $\Pi \in \mathcal{R}_{\mathcal{S}}(\mathbb{Q})$. *Under mild assumptions, if* $\mathrm{KL}(\Pi\|\mathbb{M}) < \infty$ *then we have*

$$\mathrm{KL}(\Pi\|\mathbb{M}) = \mathrm{KL}(\Pi\|\operatorname{proj}_{\mathcal{M}}(\Pi)) + \mathrm{KL}(\operatorname{proj}_{\mathcal{M}}(\Pi)\|\mathbb{M}). \tag{43}$$

*If* $\mathrm{KL}(\mathbb{M}\|\Pi) < \infty$ *then we have*

$$\mathrm{KL}(\Pi\|\mathbb{M}) = \mathrm{KL}(\Pi\|\operatorname{proj}_{\mathcal{R}_{\mathcal{S}}(\mathbb{Q})}(\Pi)) + \mathrm{KL}(\operatorname{proj}_{\mathcal{R}_{\mathcal{S}}(\mathbb{Q})}(\Pi))\|\mathbb{M}). \tag{44}$$

*Proof.* **Proof of** (43)**:** The first identity follows from algebraic manipulations of expressions for the relevant KL divergences. From the proof of Proposition 3.4, we have

$$\mathrm{KL}(\Pi\|\mathbb{M}) = \sum_{e\in\mathcal{E}} \frac{1}{2\sigma^2} \int_0^{T_e} \mathbb{E}_{\Pi^e_{0,t}}\left[\|\sigma^2\mathbb{E}_{\Pi^e_{T|0,t}}\left[\nabla\log\mathbb{Q}_{T|t}(X_T|X_t)|X_t,X_0\right] - v_e(X_t,t)\|^2\right]\mathrm{d}t \tag{45}$$

(where we suppress the superscript $e$ on the $X_t$ for notational convenience). Likewise, it can be shown that

$$\mathrm{KL}(\operatorname{proj}_{\mathcal{M}}(\Pi)\|\mathbb{M}) = \sum_{e\in\mathcal{E}} \frac{1}{2\sigma^2} \int_0^{T_e} \mathbb{E}_{X_t\sim\Pi^e_t}\left[\|\sigma^2\mathbb{E}_{\Pi^e_{T|t}}\left[\nabla\log\mathbb{Q}_{T|t}(X_T|X_t)|X_t\right] - v(X_t,t)\|^2\right]\mathrm{d}t. \tag{46}$$

From another application of the expression in Proposition 3.4, we have

$$\mathrm{KL}(\Pi\|\operatorname{proj}_{\mathcal{M}}(\Pi)) = \sum_{e\in\mathcal{E}} \frac{1}{2\sigma^2} \int_0^{T_e} \mathbb{E}_{X_0,X_t\sim\Pi^e_{0,t}}\Big[\|\sigma^2\mathbb{E}_{\Pi^e_{T|0,t}}\left[\nabla\log\mathbb{Q}_{T|t}(X_T|X_t)|X_t,X_0\right]$$

$$- \sigma^2\mathbb{E}_{\Pi^e_{T|t}}\left[\nabla\log\mathbb{Q}_{T|t}(X_T|X_t)|X_t\right]\|^2\Big]\mathrm{d}t \tag{47}$$

$$= \sum_{e\in\mathcal{E}} \frac{\sigma^2}{2} \int_0^{T_e} \mathbb{E}_{X_0,X_t\sim\Pi^e_{0,t}}\Big[\|\mathbb{E}_{\Pi^e_{T|0,t}}\left[\nabla\log\mathbb{Q}_{T|t}(X_T|X_t)|X_t,X_0\right]\|^2\Big]$$

$$- \mathbb{E}_{X_t\sim\Pi^e_t}\Big[\|\mathbb{E}_{\Pi^e_{T|t}}\left[\nabla\log\mathbb{Q}_{T|t}(X_T|X_t)|X_t\right]\|^2\Big]\mathrm{d}t, \tag{48}$$

where to obtain the second line, we have expanded out the square and taken expectations over $X_0$ in the cross-term.

Using these expressions, by applying the same algebraic manipulations as the proof of Lemma 6 in Shi et al. (2023) to each term in the summations, one obtains

$$\mathrm{KL}(\Pi\|\operatorname{proj}_{\mathcal{M}}(\Pi)) + \mathrm{KL}(\operatorname{proj}_{\mathcal{M}}(\Pi)\|\mathbb{M}) = \mathrm{KL}(\Pi\|\mathbb{M}) \tag{49}$$

as required.

**Proof of** (44)**:** The second part also follows similarly to Shi et al. (2023), but instead conditioning on the values at $\mathcal{S}$. Let $\Pi^* = \operatorname{proj}_{\mathcal{R}_{\mathcal{S}}(\mathbb{Q})}(\mathbb{P}) = \mathbb{P}_{\mathcal{S}}\mathbb{Q}_{\cdot|\mathcal{S}}$. Using the change of measure formula for KL divergence and the fact that $\Pi$ and $\Pi^*$ have the same bridges conditional on $\mathcal{S}$,

$$\mathrm{KL}(\mathbb{P}\|\Pi) = \mathrm{KL}(\mathbb{P}\|\Pi^*) + \int_{\mathcal{C}_{\mathcal{T}}} \log(\frac{\mathrm{d}\Pi^*}{\mathrm{d}\Pi}(\omega))\mathrm{d}\mathbb{P}(\omega) \tag{50}$$

$$= \mathrm{KL}(\mathbb{P}\|\Pi^*) + \int_{(\mathbb{R}^s)^{|\mathcal{S}|}} \log(\frac{\mathrm{d}\Pi^*_{\mathcal{S}}}{\mathrm{d}\Pi_{\mathcal{S}}}(x_{\mathcal{S}}))\mathrm{d}\mathbb{P}_{\mathcal{S}}(x_{\mathcal{S}}) \tag{51}$$

$$= \mathrm{KL}(\mathbb{P}\|\Pi^*) + \int_{(\mathbb{R}^s)^{|\mathcal{S}|}} \log(\frac{\mathrm{d}\Pi^*_{\mathcal{S}}}{\mathrm{d}\Pi_{\mathcal{S}}}(x_{\mathcal{S}}))\mathrm{d}\Pi^*_{\mathcal{S}}(x_{\mathcal{S}}) \tag{52}$$

$$= \mathrm{KL}(\mathbb{P}\|\Pi^*) + \mathrm{KL}(\Pi^*\|\Pi) \tag{53}$$

as required. Note that in the second line we have used the fact that $\Pi$ and $\Pi^*$ have the same bridges conditional on $\mathcal{S}$, and in the third line we have used that $\mathbb{P}_{\mathcal{S}} = \Pi^*_{\mathcal{S}}$ by construction. $\square$

**Algorithm 2:** TreeDSBM for Wasserstein barycentre computation

---

**Input:** Initial coupling $\Pi_{\mathcal{S}}^0$ over measures $\mu_i$ (e.g. independent),
        Number of IMF iterations $N$,
        Entropic regularisation parameter $\sigma$.

Construct initial reciprocal process as $\Pi^0 = \Pi_{\mathcal{S}}^0 \mathbb{Q}_{\cdot|\mathcal{S}}$. That is,

- Sample $Y_{\mathcal{S}}$ from the initial coupling $\Pi_{\mathcal{S}}^0$ over the marginals $\mu_i$;
- Sample from the unknown central marginal as $Y_0 \sim \mathcal{N}\left(\sum \lambda_i Y_i, \sigma^2 Id\right)$;
- Training samples are obtained from Brownian bridges along each edge $i$ between $Y_i$ and $Y_0$.

**for** $n \in \{0, \dots, N-1\}$ **do**

    Learn $2|\mathcal{E}|$ vector fields using bridge-matching loss (15) along each edge, using samples
    from current reciprocal process $\Pi^n$, to obtain Markovian process $\mathbb{M}^{n+1}$;

    Construct next reciprocal process $\Pi^{n+1} = \mathbb{M}_{\mathcal{S}}^{n+1} \mathbb{Q}_{\cdot|\mathcal{S}}$ using samples from $\mathbb{M}_{\mathcal{S}}^{n+1}$. That is,

- Simulate $\mathbb{M}^{n+1}$ starting from a chosen root $r \in \mathcal{S}$ (one of the known measures $\mu_i$) to obtain samples $Y_{\mathcal{S}}$ from $\mathbb{M}_{\mathcal{S}}^{n+1}$;
- Sample from the unknown central marginal as $Y_0 \sim \mathcal{N}\left(\sum \lambda_i Y_i, \sigma^2 Id\right)$;
- Training samples are obtained from Brownian bridges along each edge $i$ between $Y_i$ and $Y_0$.

**end**

---

We can finally state the result regarding convergence of the TreeIMF iterates to the TreeSB solution.

**Theorem 3.5** (**Convergence of TreeIMF**). *Under mild conditions, the* $\mathrm{TreeSB}_{\mathrm{dyn}}$ *solution* $\mathbb{P}^*$ *is the unique fixed point of the TreeIMF iterates* $\mathbb{P}^n$, *and we have* $\lim_{n\to\infty} \mathrm{KL}(\mathbb{P}^n \| \mathbb{P}) = 0$.

*Proof.* In light of the Pythagorean property in Lemma B.7, this follows from the same compactness argument as in Proposition 7 and Theorem 8 in Shi et al. (2023). $\square$

## C   Implementation details and extensions

**Algorithm for barycentre setting**   In Algorithm 2, we provide a simplified and more explicit version of Algorithm 1 for the case of computing Wasserstein barycentres.

**Illustration of TreeDSBM**   The TreeDSBM procedure for Wasserstein barycentre computation (that is, for a star-shaped tree) is illustrated in Figure 1. We also provide a diagram for a more general tree structure in Figure 4.

### C.1   Implementation details and design choices

We implement the TreeDSBM procedure using the JAX framework (Bradbury et al., 2018). Below, we outline some of the design considerations for implementing the method.

**Vector field parameterisation**   In our implementation, we use separate neural networks to parameterise the vector fields for each direction along each edge, totalling $2|\mathcal{E}|$ networks in total. Note that one could alternatively use a shared network along each edge with an additional binary input indicating the direction; this parameterisation was used in De Bortoli et al. (2024).

**Loss function**   We incorporate both the forwards and backwards losses in Equation (15) into a single loss function, and thus optimise the forward and backwards directions simultaneously. One could incorporate all edges into a single loss function and train simultaneously, or alternatively could parallelise the edge optimisations across devices because the edges are optimised independently. This is a strength of our approach and can lead to large speed-ups in training, as training time does not need to increase in proportion to the number of edges.

**Simulation**   As we train both directions along each edge, simulation of the SDEs along the tree structure can be initialised from any of the observed nodes in $\mathcal{S}$. Following Shi et al. (2023), one can rotate the starting vertex between IMF iterations. This helps to mitigate any drift that accumulates in the marginals, as the coupling samples will only use true samples from the current starting marginal; for an analysis of this, see De Bortoli et al. (2024). In our experiments, we often did

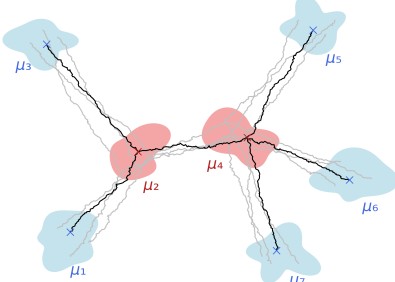
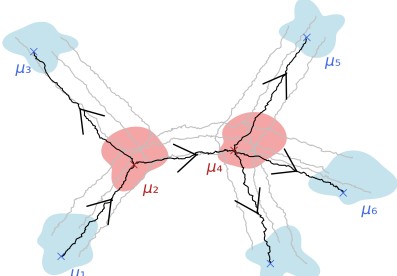

(a) Reciprocal process $\Pi$: The $Y_\mathcal{S}$ ($\times$) are sampled from the current coupling $\Pi_\mathcal{S}$ over $\mathcal{S}$. Conditional on the $Y_\mathcal{S}$, points ($\times$) at marginals $\mathcal{V}\backslash\mathcal{S}$ are sampled as $Y_{\mathcal{S}^c|\mathcal{S}} \sim \mathbb{Q}_{\mathcal{S}^c|\mathcal{S}}$. Brownian bridges are drawn along the edges between the samples.

(b) Markovianised process $\mathbb{P}$: Vector fields are trained by bridge-matching along each edge. Samples ($\times$) from the next coupling $\Pi_\mathcal{S}$ are obtained by simulating the resulting SDEs along the tree structure, started at one of the known marginals.

Figure 4: The two stages of the TreeIMF procedure, for a non-star-shaped tree structure. On this tree, the marginals at the leaf vertices $\mathcal{S} = \{1, 3, 5, 6, 7\}$ (blue) are fixed. The marginals at vertices $\mathcal{V}\backslash\mathcal{S}$ (red) are not fixed, and change during the procedure.

not observe noticeable drift in the marginals, and instead generated samples in the coupling by initialising equally across the observed vertices in $\mathcal{S}$. In our sampling, we used an Euler-Maruyama discretisation scheme with 50 uniformly-spaced steps.

**Initial coupling** Unless otherwise stated, we used the independent coupling $\Pi_\mathcal{S}^0 = \bigotimes_{i\in\mathcal{S}} \mu_i$ as the initial coupling. We note that any coupling over $\mathcal{V}$ with correct marginals on $\mathcal{S}$ could be used, and we discuss some possible alternatives in subsequent sections.

**Architectures** Other than image experiments, we use a basic MLP-based vector field model. It consists of an MLP spatial embedding with hidden layers [128, 256] to embed into dimension 32, a time embedding consisting of a sine positional-encoding and an MLP with hidden layers [128, 256] also embedding into dimension 32, before concatenating the embeddings and passing through an MLP with hidden layers [512, 256, 128]. This is the same architecture used in De Bortoli et al. (2021) and Noble et al. (2023). In image experiments, we use the UNet architecture (Ronneberger et al., 2015) with the improvements from Dhariwal and Nichol (2021), using the JAX implementation from Song et al. (2023). For all experiments, we use the Adam optimiser (Kingma and Ba, 2015) with default parameters 0.9 and 0.999.

We note that for pointcloud experiments, capacity of the neural networks is not a limiting factor—any sufficiently-expressive network will work similarly well and fairly small MLP networks suffice, so using several networks for the different edges does not pose issues. One could also use a single network across the edges and additionally condition on the edge. This makes particular sense for problems with shared structure between edges, such as those for image data (and indeed these are settings for which the networks would be larger, and maintaining multiple networks could become a computational bottleneck).

**Memory requirements** The parallel nature of the TreeDSBM algorithm during training provides practitioners with a trade-off between memory consumption and wall-clock time. Namely, one can train the edges simultaneously if compute allows (either on a single GPU if enough memory, or parallelised across GPUs). If this cannot be done, one can train sequentially instead (in which case memory requirements for training would be comparable with standard bridge-matching). We report GPU consumption for different experiments in Table 5 (for sequential and joint training), for the hyperparameters used in the paper. These can of course be changed significantly by changing hyperparameters such as batch size. We note that in some of these experiments (e.g. the Gaussian experiments), peak GPU usage is due to the simulation and storage of the training samples for subsequent IMF steps, rather than during the network training. This can be reduced significantly by simulating in smaller batches, or by updating the cache during training rather than simulating all beforehand.

Table 5: Comparing memory usage (in MB) for sequential and joint training, for the hyperparameters used.

|  | Sequential Training (MB) | Joint Training (MB) |
| --- | --- | --- |
| 2d | 447 | 687 |
| Data Aggregation (Poisson) | 705 | 1219 |
| Gaussian ($d = 64$) | 1239 | 1239 |
| MNIST 2,4,6 | 4683 | 6407 |

We remark that TreeDSBM has improved memory requirements compared to TreeDSB. In TreeDSB, training a time-reversal along an edge requires saving entire trajectories simulated along the reverse direction, whereas TreeDSBM only requires storing endpoints (this is one of the most significant benefits of IMF over IPF).

**Alternative reference measures**   We have presented the methodology according to using Brownian reference measures along each edge. However, the methodology extends to other reference process, as long as the respective bridging processes conditioned on the values at $\mathcal{S}$ are tractable (for example, this is true for Ornstein-Uhlenbeck processes). The connection to quadratic-cost optimal transport is however less simple beyond the Brownian case. See Shi et al. (2023) for a more detailed treatment of this general case.

## C.2   Methodology extensions for improving convergence speed

We now discuss possible extensions of the TreeDSBM algorithm to improve convergence speed, in the vein of existing extensions of Schrödinger bridge methodology for the standard two-marginal setting.

**Warmstarting with minibatch mmOT couplings**   The TreeIMF procedure can be initialised with any coupling $\Pi_{\mathcal{S}}^0$ over $\mathcal{S}$ with correct marginals. For the standard SB problem, Tong et al. (2024b) note that the SB solution is a mixture of bridges mixed by a static $\varepsilon$-OT solution, and thus propose a single iteration of bridge matching on samples generated by a static $\varepsilon$-OT solver applied on minibatches (see also Pooladian et al. (2023), Tong et al. (2024a), and Fatras et al. (2021)).

Similarly, one can initialise TreeIMF using samples obtained from static mmOT solvers applied to minibatches. Such a procedure can speed up convergence to the TreeSB solution by initialising closer to the true solution. We remark, however, that such minibatching approaches can incur large errors (particularly in higher dimensions or for small minibatches; for example the Wasserstein-1 error grows as $O(B^{-1/(2d)})$ (Sommerfeld et al., 2019)), and so the advantages of such methods are lessened as dimensionality increases.

**Flow-based IMF on the tree**   Iterative Markovian Fitting presents a mathematically elegant approach for solving the SB problem, with significant practical improvements over the IPF procedure. However, the iterative nature of the algorithm remains a downside - each iteration of the two-step procedure involves first simulating the current Markovian process, and then retraining a neural network with the bridge-matching loss. In practice, one might wonder if it is possible to perform the simulations and bridge-matching procedure simultaneously to avoid the expensive iterative nature of the algorithm. The recent work of De Bortoli et al. (2024) answered this in the affirmative and propose the $\alpha$-IMF procedure, which instead corresponds to a discretisation of a *continuous flow* of processes that converge to the SB. We anticipate one could design an analogous methodology for the tree-based setting that we consider; we leave such extensions for future work.

# D   Experimental details

Here, we provide experimental details and additional results and discussion regarding the experiments included in the main body.

## D.1   Synthetic $2d$ barycentre

**Datasets**   We follow the experimental setup of Noble et al. (2023). The marginals consist of moon, circle, and spiral datasets from scikit-learn (Pedregosa et al., 2011), centred and scaled by a factor of 7.0. We aim to learn the $(\frac{1}{3}, \frac{1}{3}, \frac{1}{3})$-barycentre of the dataset. This is a challenging problem—the

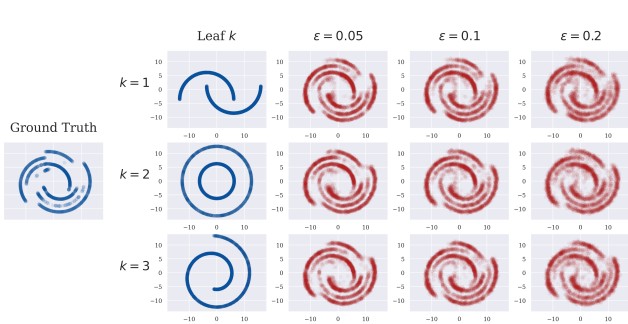

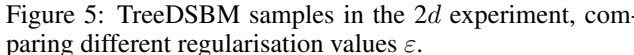

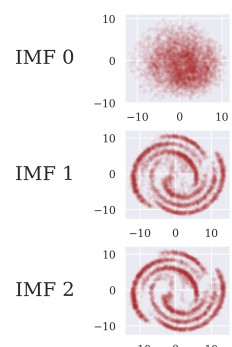

Figure 5: TreeDSBM samples in the $2d$ experiment, comparing different regularisation values $\varepsilon$.

Figure 6: Progression of TreeDSBM ($\varepsilon = 0.1$) barycentre approximations through the IMF iterations.

Table 6: Progression of the Sinkhorn divergence to the ground truth, and the average Sinkhorn divergence to the marginals, during the IMF iterations (mean±std, over 5 runs).

|  | IMF 1 | IMF 2 | IMF 3 | IMF 4 | IMF 5 | IMF 6 |
|---|---|---|---|---|---|---|
| Fit to solution | $17.7 \pm 0.2$ | $1.28 \pm 0.04$ | $1.21 \pm 0.07$ | $1.13 \pm 0.05$ | $1.12 \pm 0.05$ | $1.09 \pm 0.06$ |
| Fit to marginals | $0.16 \pm 0.03$ | $0.17 \pm 0.03$ | $0.18 \pm 0.04$ | $0.19 \pm 0.03$ | $0.19 \pm 0.03$ | $0.20 \pm 0.02$ |

lower-dimensional and discontinuous structures in the marginals mean that the barycentre has a complex and fragmented support, and the transport maps to the barycentre are highly discontinuous.

**Hyperparameters**  For TreeDSBM, we use $\varepsilon = 0.1$ and run for 6 IMF iterations. For training the vector fields, we use 10,000 training steps and a batch size of 4096. We use the Adam optimiser (with default parameters 0.9, 0.999) with learning rate 1e-3 and exponential moving average parameter of 0.99. At inference we use the Euler-Maruyama scheme with 50 steps. We generate a batch of 10,000 training couplings for subsequent TreeIMF iterations (a third simulated from each marginal). For the other algorithms, we use their default parameters provided in their respective codebases.

**Comparison with alternative algorithms**  For the ground-truth, we use the in-sample free-support barycentre algorithm of Cuturi and Doucet (2014) implemented in Python Optimal Transport (Flamary et al., 2021), with 1500 datapoints. Note that the aim of this experiment is not to outperform in-sample methods; it is known that such approaches perform well in low dimensions, but do not scale well as dimension increases. The in-sample method is used here to provide a close approximation to the ground-truth, allowing us to judge the success of the continuous Wasserstein-2 barycentre approaches that we compare.

For TreeDSB, we use the checkpoints provided by Noble et al. (2023) which were trained for 50 IPF iterations.

We also report results for the WIN algorithm from Korotin et al. (2022). This is an iterative algorithm inspired by Álvarez-Esteban et al. (2016); it pushes forward a source latent distribution $\rho$ through a function $G$ to give a generative model $\nu = G\#\rho$ for the barycentre, and also learns maps $T_i$ and $T_i^{-1}$ transporting from the generated barycentre to and from the marginals respectively. In our experiments, the barycentre generator $\nu = G\#\rho$ was unable to fit the true barycentre accurately, nor were the maps $T_i^{-1}\#\mu_i$. We hypothesise that the neural maps struggle to model the discontinuous transports well. However, the combined map $(\sum_i \lambda_i T_i)\#\nu$ was able to give a good approximation of the true barycentre, which are the results we report in Figure 2 and Table 2.

Additionally, we applied the W2CB (Korotin et al., 2021) and NOTWB (Kolesov et al., 2024a) algorithms to this example, but were unable find hyperparameters to make the algorithms to converge to the correct solution. Again, this may be due to the difficulty in modelling the discontinuous transports with neural networks. We also anticipate that the loss landscapes caused by this complex

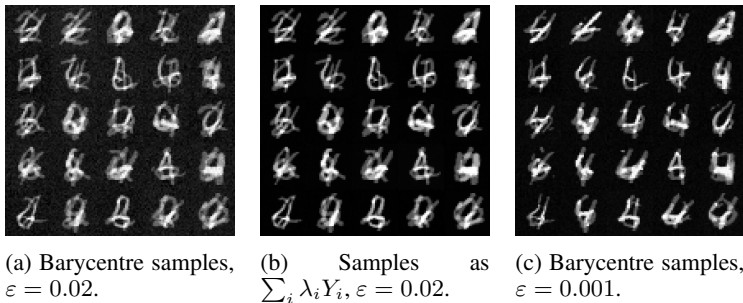

(a) Barycentre samples, $\varepsilon = 0.02$.

(b) Samples as $\sum_i \lambda_i Y_i$, $\varepsilon = 0.02$.

(c) Barycentre samples, $\varepsilon = 0.001$.

Figure 7: (a) The true TreeSB solution will have noise present due to entropy-regularisation. If one wishes to reduce this, one can instead (b) construct samples as $\sum_i \lambda_i Y_i$, or (c) use a smaller $\varepsilon$ value.

example may have caused these methods to get stuck in local minima. It appears that the iterative schemes of TreeDSBM, TreeDSB, and WIN aid in overcoming such issues.

**Convergence speed**   In Figure 6, we demonstrate the progression of the TreeDSBM barycentre approximation as we run the IMF iterations. We run with $\varepsilon = 0.1$, and plot samples generated from leaf 0. In Table 6, we also show how the Sinkhorn divergence to the ground truth evolves as the IMF iterations progress. After only two IMF iterations, TreeDSBM already gives a good approximation to the solution. Such behaviour reflects similar results reported in Lindheim (2023), which observes that iterative fixed-point approaches (Álvarez-Esteban et al., 2016) exhibit very fast convergence to the solution. TreeDSBM performs only a single bridge-matching iteration along each edge before updating the barycentre, rather than computing full OT maps (which would be expensive). As such, it strikes a good balance between the efficiency of iterative fixed-point-based approaches, without requiring full OT map computations before updating the barycentre approximation.

**Runtime analysis**   We report approximate runtimes for the three methods that converged. All experiments were ran on a single Nvidia GeForce RTX 2080Ti GPU.

Our TreeDSBM implementation took approximately 1 minute for each IMF iteration when training the edges jointly, and took around 7 minutes to run the 6 IMF iterations. Note that one could also obtain good results using fewer IMF iterations or fewer training steps.

In contrast, the alternative methods were significantly slower. WIN required around 8000 training steps to obtain a good barycentre approximation, which took approximately 1 hour 20 minutes. The provided checkpoint for TreeDSB is for 50 IPF iterations, each of which would require 6 time-reversal training procedures, and would thus take significantly longer to train than TreeDSBM.

**Additional results**   In Figure 5, we plot TreeDSBM samples obtained for different values of entropy-regularisation $\varepsilon$. As expected, increasing $\varepsilon$ leads to a slight blurring bias in the solution. In Table 6 we also report the quality of fit to the marginals, calculated by simulating from the 'moon' marginal to the centre and then out to the other leaf nodes, and averaging the resulting Sinkhorn divergences to ground-truth samples from these marginals. We see that in this experiment there is negligible drift accumulation in the marginals.

We overall found TreeDSBM to perform strongest in this experiment. Its bridge-matching objectives and iterative nature provide fast and stable training in what is a complex and challenging problem setting, and its dynamic-transport approach means that it is able to model the discontinuous transport maps accurately.

## D.2   MNIST 2,4,6 barycentre

We also follow the experimental setup of Noble et al. (2023) and compute the $(\frac{1}{3}, \frac{1}{3}, \frac{1}{3})$-barycentre of the 2,4 and 6 digits in the MNIST dataset (LeCun et al., 2010).

**Hyperparameters**   We use the UNet architecture of Song et al. (2023), with 64 channels, channel multiples of (1,2,2), attention at layers (16,8), and 2 residual blocks at each layer. We train each bridge-matching procedure with 10,000 steps with batch size 64, at a learning rate of 1e-4 and with

exponential moving average weight of 0.999. For subsequent IMF iterations, we simulate 8,192 coupling samples. We run for 4 IMF iterations, beyond which we did not see much change between iterations. The TreeDSB samples plotted in Section 4 are those displayed in Noble et al. (2023), as we did not have the computational resources to run TreeDSB to convergence in this setting.

**Role of** $\varepsilon$   In Section 4, the TreeDSBM barycentre has some noise present in the samples. We remark here that this noise *should* be present in an accurately computed solution, as we are solving for the $\varepsilon$-TreeSB solution which adds entropic regularisation. If one wishes for less noise in the samples, one can run with a smaller entropy-regularisation value $\varepsilon$. Alternatively, one could generate samples by sampling $Y_i$ according to the learned coupling, and weighting them as $\sum_i \lambda_i Y_i$ (though note that would not be clear what kind of barycentre such samples would be from). We plot results for a smaller $\varepsilon = 0.001$, and for the weighted coupling samples $\sum_i \lambda_i Y_i$ for $\varepsilon = 0.02$ in Figure 7, to demonstrate that TreeDSBM can generate samples with minimal noise if so desired.

**Convergence speed**   We provide an analysis of the convergence speed of TreeDSBM in terms of the number of IMF iterations required. Unfortunately there is no ground-truth barycentre to compare to in this example; this makes quantitative evaluation of the obtained barycentre difficult. We therefore instead assess the fit to the marginals (from which we have true samples) as a proxy for the success of the algorithm, along with the transport cost — the transport cost provides an indication of the optimality of the maps, while if the marginals are not fitted accurately, then the resulting barycentre will be unreliable. While not an ideal measure of the 'quality' of the barycentre itself, this does provide a quantitative and, importantly, tractable proxy for the 'success' of the algorithm.

To this end, we report in Table 7 the transport cost and FID values for samples from the marginals, for the 4 IMF iterations (note that we train a classifier and use the obtained features for the FID calculation, so these values should not be compared with those in other works). We initialise the sampling from 1000 unseen test samples of the digit 6, and report the FID values of the obtained 2s and 4s (averaged). We see that, as expected, the transport cost decreases as the IMF iterations proceed, indicating that the barycentre approximation is improving. We also observe that the FID scores increase slightly (though there is little visible difference) — this is a consequence of the drift that can accrue in the marginals, and is consistent with the expected behaviour of DSBM from which this effect is inherited (this can be reduced by training for longer or using the techniques discussed in Appendix C.1).

Table 7: Progression of the total transport cost and fit to the marginals (as measured by FID), during the IMF iterations. Note that the FID values are obtained using a trained classifier, so should not be compared to values in other works.

|  | IMF 1 | IMF 2 | IMF 3 | IMF 4 |
|---|---|---|---|---|
| Transport cost | 431 | 402 | 389 | 378 |
| Ave. FID (2s and 4s) | 61 | 82 | 89 | 93 |

### D.3   Subset posterior aggregation

The previous experiments have shown TreeDSBM to improve over its IPF counterpart TreeDSB. In the following experiments, we provide a more detailed comparison with current strongly-performing methods for continuous Wasserstein-2 barycentre estimation, by reporting results for standard experiments in the literature. We include comparisons against the methods WIN (Korotin et al., 2022), W2CB (Korotin et al., 2021), and the recent method NOTWB (Kolesov et al., 2024a). These methods have demonstrated strong empirical performance in their respective works, and are chosen here to be representative of different approaches in the literature—WIN is an iterative method inspired by Álvarez-Esteban et al. (2016), W2CB is an Input Convex Neural Network-based approach (Amos et al., 2017; Makkuva et al., 2020), and NOTWB is based on recent Neural OT methodology (Korotin et al., 2023). We use the implementations in their publicly available code, to which we provide the links in Section F. We note that, as ever in barycentre studies, it is somewhat challenging to assess the performance of solvers due to the lack of the ground-truth solution (other than in certain specific examples). Here, we report results on standard experiments used in the literature.

**Experimental setup** We work with the experimental setup and dataset used in Korotin et al. (2021) (the same dataset was also used previously in Li et al. (2020) and Fan et al. (2021)), which uses Poisson and negative-binomial regressions on a bike-rental dataset (Fanaee-T, 2013). The aim is to predict the hourly number of bike rentals using features including day of the week, weather conditions, and more. The dataset is 8-dimensional and is split into 5 distinct subsets each of size 100,000. The 'ground-truth' barycentre consists of 100,000 samples from the full dataset posterior.

Following the literature, we report the $\mathrm{BW}_2^2-\mathrm{UVP}$ metric between the 'ground-truth' and the obtained samples in Table 3. For methods that generate from each marginal, we report the average over generations from each marginal, and for WIN we report results for the barycentre generator. The $\mathrm{BW}_2^2-\mathrm{UVP}$ metric is defined as

$$\mathrm{BW}_2^2-\mathrm{UVP}(\nu, \tilde{\nu}) = 100 \cdot \frac{\mathrm{BW}_2^2(\nu, \tilde{\nu})}{\frac{1}{2}\mathrm{Var}(\tilde{\nu})}\%, \tag{54}$$

where the Bures-Wasserstein metric is defined as $\mathrm{BW}_2^2(\nu, \tilde{\nu}) = W_2^2(\mathcal{N}(m_\nu, \Sigma_\nu), \mathcal{N}(m_{\tilde{\nu}}, \Sigma_{\tilde{\nu}}))$ for the respective means and covariances of the distributions.

**Hyperparameters** For TreeDSBM, we use $\varepsilon = 0.001$ and run for 4 IMF iterations. For training the vector fields, we use 2000 training steps and a batch size of 4096. We use the Adam optimiser (with default parameters 0.9, 0.999) with learning rate 1e-3 and exponential moving average parameter of 0.99. At inference we use the Euler-Maruyama scheme with 50 steps. We generate a batch of 50,000 training couplings for subsequent TreeIMF iterations (10,000 simulated from each marginal).

For W2CB, we use a learning rate of 1e-4 in the negative binomial setting. For WIN, in the negative binomial case we rescale the source $z$-sampler by a factor of 10.0 to match the scale of the data better; without this, it did not appear to converge. We run W2CB and WIN for 10000 training iterations, and NOTWB for 2500 iterations. Other than those mentioned, we use the default parameters provided in the respective codebases.

**Runtime analysis:** We report approximate runtimes for the different approaches; all experiments were ran on a single Nvidia GeForce RTX 2080Ti GPU. To compare approximate time taken, we report time taken to for the methods to converge close to their final output - chosen by monitoring the $\mathrm{BW}_2^2-\mathrm{UVP}$ metric and choosing the time beyond which it no longer decreases significantly (note these are not the amount time used in Table 3, which we trained using the hyperparameters described above). This is somewhat subjective, but is a fairer comparison than just reporting times for running with default parameters. We remark that it may be possible to improve these runtimes with further hyperparameter tuning and by optimising the algorithm implementations, but investigating such optimisations is beyond the scope of this work.

The W2CB algorithm appeared to give good results after approximately 1000 training steps in both cases, which took around 10 minutes in our experiments. WIN converged after around 2500 iterations, which took around 45 minutes. NOTWB converged quickly after only around 200 iterations, which took around 2 minutes.

In both experiments, each TreeIMF iteration for our TreeDSBM implementation took approximately 20 seconds when training the edges jointly. TreeDSBM converged well using 4 IMF iterations, and training took around 2 minutes 30 seconds (including time for simulating training samples for the next iteration). This is comparable with NOTWB, the fastest of the alternative methods.

**Convergence speed** We provide the values of the $\mathrm{BW}_2^2-\mathrm{UVP}$ metric as the IMF iterations progress in Table 8, and again we observe very fast convergence.

Table 8: Progression of the $\mathrm{BW}_2^2-\mathrm{UVP}$ metric during the IMF iterations, for the subset posterior aggregation experiment (mean±std, over 5 runs).

|  | IMF 1 | IMF 2 | IMF 3 | IMF 4 |
|---|---|---|---|---|
| ↓ Poisson | $31.1 \pm 0.03$ | $0.0085 \pm 0.0003$ | $0.0075 \pm 0.0006$ | $0.0076 \pm 0.0005$ |
| ↓ Negative Binomial | $31.0 \pm 0.01$ | $0.0123 \pm 0.0003$ | $0.0118 \pm 0.0007$ | $0.0121 \pm 0.0004$ |

## D.4 Higher-dimensional Gaussian experiments

**Experimental setup** We follow the experimental setup previously used in (Korotin et al., 2021, 2022; Kolesov et al., 2024a), in which 3 Gaussian distributions and its ground-truth $(\frac{1}{3}, \frac{1}{3}, \frac{1}{3})$-barycentre are randomly generated, for each dimension in $\{64, 96, 128\}$. We report results for $\mathrm{BW}_2^2-\mathrm{UVP}$ (which measures the quality of the overall generated barycentre) as described above, and additionally report the $L^2-\mathrm{UVP}$ metric, which measures the quality of the individual maps to the barycentre and is defined for each marginal as

$$L^2-\mathrm{UVP}(\hat{T}, T^*) = 100 \cdot \frac{\|\hat{T} - T^*\|^2}{\mathrm{Var}(\tilde{\nu})}\%, \tag{55}$$

where $\hat{T}$ denotes the learned map from the marginal to the barycentre, and $T^*$ is the known ground truth mapping. Again, we provide the averages over the marginals.

**Hyperparameters** For TreeDSBM, we use $\varepsilon =$1e-4 and run for 4 IMF iterations. For training the vector fields, we use 10,000 training steps and a batch size of 4096. We use learning rate 1e-3 and exponential moving average parameter of 0.99. We generate a batch of 50,000 training couplings for subsequent TreeIMF iterations (simulated equally from each marginal). Results reported for TreeDSBM are the average over 5 runs.

For the alternative methods, we run W2CB with learning rate 1e-4, and otherwise use the default parameters provided in their respective codebases.

**Convergence speed** We provide the values of the $\mathrm{BW}_2^2-\mathrm{UVP}$ metric as the IMF iterations progress in Table 8, and again we observe very fast convergence.

Table 9: Progression of the $\mathrm{BW}_2^2-\mathrm{UVP}$ and $L^2-\mathrm{UVP}$ metrics during the IMF iterations, for the Gaussian $d = 64$ experiment (mean±std, over 5 runs).

|  | IMF 1 | IMF 2 | IMF 3 | IMF 4 |
|---|---|---|---|---|
| $\downarrow \mathrm{BW}_2^2-\mathrm{UVP}$ | $16.0 \pm 0.01$ | $0.12 \pm 0.01$ | $0.13 \pm 0.02$ | $0.14 \pm 0.03$ |
| $\downarrow L^2-\mathrm{UVP}$ | $16.7 \pm 0.01$ | $1.19 \pm 0.02$ | $1.18 \pm 0.02$ | $1.18 \pm 0.03$ |

**Discussion of continuous Wasserstein-2 barycentre solver comparisons**

Our experiments show that our TreeDSBM algorithm exhibits strong performance, and is competitive against state-of-the-art methods for continuous Wasserstein-2 barycentre estimation in a range of settings. In particular, TreeDSBM offers *fast* and *stable* training—even in complex settings—due to its bridge-matching loss objectives, and comes with a well-understood theoretical analysis.

The best choice of barycentre algorithm may depend on the specific problem setting. For example, if the transport maps exhibit complex behaviour (possibly due to lower-dimensional, manifold-like structures in the datasets) then the flow-based approach of TreeDSBM will likely perform strongly (such as in the $2d$ barycentre experiment in Section 4). Also, when fast training is required then our experiments suggest TreeDSBM is a strong option. On the other hand, if fast inference is important then a one-step-generation solver such as NOTWB might be preferable. Note that one could incorporate distillation techniques from the flow-matching literature for improving the speed of TreeDSBM inference after training.

Overall, TreeDSBM offers a compelling new addition to the taxonomy of continuous Wasserstein-2 barycentre solvers, with distinctly different characteristics to alternative approaches due to its flow-based nature.

# E Additional experiments

## E.1 Further comments regarding computational considerations

**Choice of entropy regularisation** Choosing the entropy regularisation parameter $\varepsilon$ is a perennial question in entropic OT. Standard methods to choose this value in commonly used OT libraries (for example, choosing in proportion to the costs) provide good guidance for choosing suitable

Table 10: Effect of entropy-regularisation parameter $\varepsilon$ in the $2d$ and data aggregation experiments.

| | $\varepsilon = 1.0$ | $\varepsilon = 0.3$ | $\varepsilon = 0.1$ |
|---|---|---|---|
| $2d$, Sinkhorn-divergence | 1.24 | 0.99 | 1.02 |
| | $\varepsilon =$1e-3 | $\varepsilon =$3e-4 | $\varepsilon =$1e-4 |
| Data Aggregation (Poisson), $\mathrm{BW}_2^2-\mathrm{UVP}$ | 0.012 | 0.008 | 0.008 |

Table 11: Effect of batch size in the $2d$ and data aggregation experiments.

| **Batch size** | **64** | **246** | **1024** | **4096** |
|---|---|---|---|---|
| $2d$, Sinkhorn-divergence | 1.57 | 1.24 | 1.04 | 1.04 |
| Data Aggregation (Poisson), $\mathrm{BW}_2^2-\mathrm{UVP}$ | 0.032 | 0.017 | 0.013 | 0.012 |

values. Typically, one may want to choose $\varepsilon$ as small as possible the reduce the entropic bias. One advantage of TreeDSBM over TreeDSB is that it allows for much smaller epsilon (TreeDSB does not converge for too-small $\varepsilon$, as simulated trajectories struggle reach the other marginals). We provide a visualisation of the role of $\varepsilon$ in the $2d$ example in Figure 5, and also for two values of $\varepsilon$ for the MNIST experiment in Figure 7. In Table 10, we also add some further quantitative results for different $\varepsilon$ values, in the $2d$ and subset posterior aggregation settings.

**Fitting to the marginals** One of the limitations of our approach is that errors can accumulate in the marginals as IMF iterations proceed; this is a limitation inherited from standard IMF and similar reflow methods. Standard techniques from the literature can be used to mitigate this (such as rotating the starting marginal as in Shi et al. (2023), or using the projection methods in Kim et al. (2025)). It is therefore important that the bridge-matching steps fit the marginals accurately, and hyperparameters should be chosen accordingly. To provide an indication of how the learned bridge-matching quality affects the overall solution, we provide results for varying batch size on the $2d$ and data aggregation experiments in Table 11, for the hyperparameters used in the paper. We have also provided results assessing how the fit to the marginals changes as the IMF iterations progress in the $2d$ and MNIST experiments in Tables 6 and 7 respectively.

### E.2 Ave! Celeba benchmark

In this section, we provide an example that illustrates a potential limitation of our approach. As previously discussed, it is difficult to evaluate performance of barycentre algorithms in high dimensions due to the lack of a ground-truth. To combat this, Korotin et al. (2022) proposed the Ave, celeba! barycenter benchmark, which consists of 3 distributions of transformed CelebA faces (Liu et al., 2015), for which the $(\frac{1}{4}, \frac{1}{2}, \frac{1}{4})$-Wasserstein-2 barycentre recovers the true CelebA dataset. The resulting dataset consists of around 67k samples in each marginal, and each image is shape $64 \times 64 \times 3$.

We consider applying the TreeDSBM in this example. It is known that performing bridge-matching between complex datasets such as images can be challenging, so for the first step we instead pre-train models using single bridge-matching iteration from a standard Gaussian to each marginal. To obtain the next coupling for training, we run the process from one of of the marginals to the latent representation in the Gaussian, and then out to the other marginals. This aids in learning, as there is often good structure preserved between the obtained samples from each marginal. For subsequent iterations we also warmstart the parameters from these pretrained models. The experiments were conducted on Nvidia A100 GPUs on Google Colaboratory.

**Hyperparameters** We use the UNet architecture of Song et al. (2023), with 128 channels, channel multiples of (1,2,2,2), attention at layers (32,16,8), and 4 residual blocks at each layer.

We use $\sigma = 0.01$ and train each bridge-matching procedure with batch size 32, at a learning rate of 1e-4 and with exponential moving average weight of 0.999. For pretraining, we run for 20,000 training steps (which takes approximately 5 hours), and for subsequent IMF iterations we run for 10,000 steps (which each take around 2.5 hours).

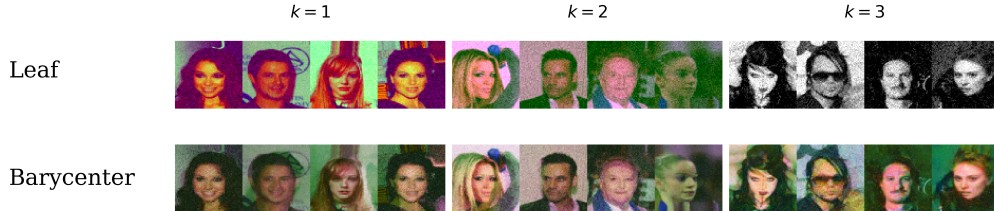

Figure 8: Samples from TreeDSBM applied to the Ave! Celeba benchmark. Many samples are transported well, but some pick up unwanted artifacts. We discuss the findings from this experiment below.

For training subsequent IMF iterations along each edge, we generate samples from the coupling from each datapoint in the corresponding marginal. This mitigates drift in the sample quality at the marginals, as we always use true datapoints from the marginal during training. We run for 2 IMF iterations, and did not see much change for subsequent IMF iterations beyond this.

**Discussion of results** TreeDSBM is able to scale to the high-dimensional setting, but the obtained samples do not match the visual quality of state-of-the-art results such as those reported in Kolesov et al. (2024a). Observe that some of the generated samples are good, but some contain additional artifacts that should not be present. We anticipate that this is due to the initial pretraining coupling. When generating samples $Y_i$ from the pretraining coupling, we simulate from one marginal to the Gaussian latent, and then out to the other marginals. This often results in strong structural similarities between the obtained $Y_i$ which yields good barycentre samples for training the next IMF iteration. However, sometimes the coupling samples $Y_i$ do not resemble each other, and the resulting barycentre sample consists of separate overlaid images. This appears to be difficult for the algorithm to recover from, resulting in the artifacts visible in some of the generated images.

Overall this suggests a limitation of TreeDSBM for this particular benchmark—the true transport maps are in fact very simple in this specific example (primarily just colour changes), but it is difficult for TreeDSBM to learn this because communication between the edges is infrequent and only occurs after each IMF iteration. In contrast, state-of-the-art methods for this benchmark optimise all the maps together and with much more interaction between them, which we anticipate is a better inductive bias for the shared structure present in this benchmark. Note that the fact that TreeDSBM optimises the edges separately is in fact a *strength* of the approach in many settings; it results in stable training without needing adversarial objectives, and allows for speed-ups by training the edges simultaneously. However, this experiment suggests that this may be a limitation of our method in scenarios where the true maps exhibit a lot of shared structure (as in the case in this example), as communication between edges occurs too infrequently to recognise this shared structure.

We remark that we observed improved performance in this benchmark by using a shared architecture over the edges (conditioning on the edge and the direction), compared to using a different network along each edge. Such an architecture makes sense in this example, as there are shared features along the edges that the network can learn and this reduces the computational and memory cost. When using a shared architecture, the network also appeared to create more consistent samples from the initial coupling. However, there are still unwanted artifacts present in many of the generated samples, and so alternative methods such as Kolesov et al. (2024a) are likely more suitable for settings such as these, as discussed above.

We anticipate that the performance of TreeDSBM in this setting could be improved through architectural changes and other implementation tricks from the flow-matching literature (for example, using preconditioned flow parameterisations (Karras et al., 2022), and techniques to mitigate marginal drift in reflow methods (Kim et al., 2025)). Such investigations offer promising directions for future work.

### E.3 Beyond star-shaped trees

So far, we have demonstrated the empirical performance of TreeDSBM only on star-shaped trees, as we have focused on computing Wasserstein barycentres. Finally, we demonstrate that our TreeDSBM also works for non-star shaped trees, and thus has potential applications beyond only barycentre computation. We consider a simple 2-dimensional example with the same tree structure

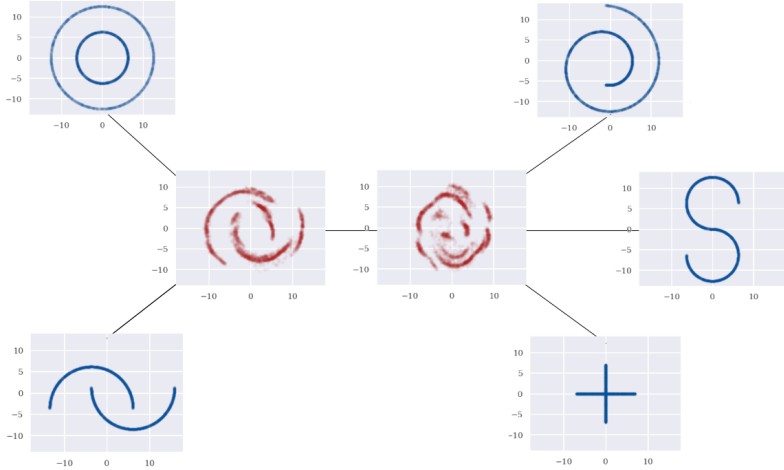

Figure 9: TreeDSBM applied to a non-star-shaped tree.

as shown in the TreeIMF diagram in Figure 4, with standard scikit-learn distributions on the observed leaves and each edge having length 1. As in the previous 2-dimensional barycentre problem, this is a challenging task due to the discontinuous transport maps and fragmented supports at the solution. We run TreeDSBM for 4 IMF iterations with $\varepsilon = 0.1$, and train each edge for 10,000 iterations with learning rate 1e-3 and exponential moving average parameter of 0.99. We plot the obtained measures in Figure 9, and see that TreeDSBM is again able to learn the complex mappings required for this setting. While direct applications of non-star-shaped trees are less clear than in the barycentre case, examples have been studied in Haasler et al. (2021) and Solomon et al. (2015), and they could have potential applications for modelling temporal behaviour of population dynamics, for example if populations were known to split according to a known structure. We leave investigating possible applications of general tree-structured costs for future work.

## F   Licenses

The following assets were used in this work.

- TreeDSB (Noble et al., 2023), MIT License
  `https://github.com/maxencenoble/tree-diffusion-schrodinger-bridge`

- WIN, Ave! Celeba dataset (Korotin et al., 2022), MIT License
  `https://github.com/iamalexkorotin/WassersteinIterativeNetworks`

- W2CB (Korotin et al., 2021), MIT License
  `https://github.com/iamalexkorotin/Wasserstein2Barycenters`

- NOTWB (Kolesov et al., 2024a), MIT License
  `https://github.com/justkolesov/NOTBarycenters`

- JAX Consistency Models (Song et al., 2023), Apache-2.0 License
  `https://github.com/openai/consistency_models_cifar10`

- Bike Sharing, UCI Machine Learning Repository (Fanaee-T, 2013), CC BY 4.0 License
  `https://archive.ics.uci.edu/dataset/275/bike+sharing+dataset`

- MNIST digits classification dataset (LeCun et al., 2010), CC BY-SA 3.0 License

- OTT-JAX (Cuturi et al., 2022), Apache-2.0 License

- Python Optimal Transport (Flamary et al., 2021), MIT License

