# OpenReview forum: "Schrödinger Bridge Matching for Tree-Structured Costs and Entropic Wasserstein Barycentres"
_NeurIPS.cc/2025/Conference — NeurIPS 2025 poster_

### Official Review · Reviewer_ZjRa · 2025-07-01

**Clarity:** 3
**Significance:** 4
**Originality:** 4
**Rating:** 5
**Confidence:** 3

**Summary:**

This paper proposes TreeDSBM, a novel algorithm that extends the Iterative Markovian Fitting (IMF) procedure to solve Schrödinger Bridge (SB) problems defined over arbitrary tree-structured cost graphs. This generalization enables scalable solutions to multi-marginal entropy-regularized optimal transport problems with structured cost functions, such as those arising in Wasserstein barycentre computation. The authors provide a rigorous theoretical foundation for TreeDSBM. They also demonstrate empirical benefits of TreeDSBM over its IPF-based counterpart, TreeDSB, in selected barycentre estimation tasks.

**Questions:**

1. The proposed method seems applicable to any multi-marginal OT problem with tree-structured costs. Can the authors provide one or two additional use cases beyond barycentre computation to demonstrate this generality?

2. It would be very helpful to compare against more recent continuous barycentre methods. Can the authors either expand the baseline set or justify their omission?

3. Could the authors report training time, inference time, and GPU memory consumption for TreeDSBM versus TreeDSB or other methods? This would help in understanding the method’s practical viability.

4. Was it inadvertently omitted in the submission?

**Ethical Concerns:**

["NO or VERY MINOR ethics concerns only"]

**Final Justification:**

Thank authors for their rebuttal and discussion. In particular, they have made a substantial effort to address the reviewers’ concerns regarding the quantitative evaluation. I recommend acceptance.

**Limitations:**

Yes

**Paper Formatting Concerns:**

No major formatting issues were found.

**Quality:**

3

**Strengths And Weaknesses:**

Strengths:

1. The extension of IMF to general tree-structured costs is a clear and non-trivial innovation. Prior flow-based SB methods have largely been restricted to two-marginal settings or relied on IPF-like iterations. This work addresses an important gap and is a natural yet challenging generalization.

2. The paper is rigorous and mathematically well-grounded. Key theoretical constructs such as KL decompositions on trees, convergence guarantees, and generalized projections are carefully introduced and justified.

Weaknesses:

1. Despite the general applicability of tree-structured cost models, the paper only evaluates TreeDSBM in the context of Wasserstein barycentre estimation. While this is a canonical and well-motivated use case, it raises questions about the method's general utility in broader machine learning applications such as structured generative modeling, domain adaptation, or trajectory inference.

2. In the barycentre setting, the paper compares TreeDSBM only against TreeDSB and WIN. The lack of broader comparisons makes it harder to contextualize the method’s empirical competitiveness.

3. A practical appeal of IMF over IPF is improved sample efficiency and training stability. However, the paper does not report computational time, memory usage, or wall-clock comparisons, which are critical for assessing scalability in high-dimensional or large-tree settings.

4. Many technical discussions, proofs, and implementation details are deferred to the appendix, but the appendix seems to be missing. This makes it difficult to fully verify the claims or reproduce the results.

---

> ### Author Rebuttal · Authors · 2025-07-30
>
> We thank the reviewer for their positive and constructive comments, and we address their comments and questions below. We first provide a **brief rebuttal summary**:
> - Appendix: The appendix is located in the zip folder under ‘Supplementary Material’: Then go `neurips_treeIMF_supplementary` -> `supplementary.pdf`. Please let us know if you have issues accessing it.
> - The Appendix contains many additional experiments and discussions relevant to the reviewer's questions, in particular comparisons with alternative continuous barycentre algorithms, and discussion of runtimes.
> - We provide additional information and discussion regarding computational resources in this rebuttal.
>
> We expand on these points in more detail below, and highlight any proposed changes for revisions in **bold**.
>
> ## Comparison with other continuous barycentre methods
>
> We do in fact provide comparisons with other continuous Wasserstein-2 barycentre solvers in the Appendix. Namely, we include comparisons with state-of-the-art methods WIN [1], W2CB [2] and NOTWB [3]. These methods are chosen to be representative of different approaches in the literature— WIN is an iterative fixed-point approach, W2CB is an Input Convex Neural Network-based method, and NOTWB is based on recent Neural OT methodology.
>
> In particular, we include in Appendix E comparisons on a subset posterior aggregation experiment (used as a standard experiment in continuous Wasserstein-2 barycentre approaches, e.g. in [2],[4]), and also on high-dimensional Gaussian examples for $d=64,96,128$ (again a standard experiment in the literature, e.g. in [1],[2],[3]). In lines 1130-1145 we provide a discussion regarding the findings, highlighting the pros and cons of different approaches in the literature.
>
> Importantly, if accepted **we intend to move the data aggregation and high-dimensional Gaussian experiments to the main paper**, using the extra page afforded to us.
>
>
> ## Training time, inference time, GPU memory consumption
>
> Training time:
> - We discuss run-times of the various algorithms that we compare in the Appendix in lines 941-952, and in lines 1079-1095, and TreeDSBM appeared favourable compared to existing barycentre solvers.
>
> Inference time:
> - Inference time for TreeDSBM is slower than for methods that use a single function evaluation, as it involves solving an SDE (discussed in line 361 in main paper, and line 1139 in the Appendix). However, it can still be fast if one JIT-compiles the sampling procedure: transporting 1,000 samples along one of the maps in the data aggregation experiment takes 7.8 ms for the single-evaluation NOTWB method, and 15 ms for TreeDSBM when using 50 discretisation steps.
>
> GPU memory consumption
> - We comment here that the parallel nature of the algorithm during training means that a practitioner can have a trade-off between memory consumption and wall-clock time. Namely, one can train the edges simultaneously if compute allows (either on a single GPU if enough memory, or parallelised across GPUs). If this cannot be done, one can train sequentially instead (in which case memory requirements for training would be comparable with standard bridge-matching).
> - We report GPU consumption for different experiments below (for sequential and joint training), for the hyperparameters used in the paper. These can be changed significantly by changing hyperparameters such as batch size.
>
> | Experiment                 | Sequential Training | Joint Training  |
> | -------------------------- | ---------- | ------- |
> | 2d                         | 447 MB     | 687 MB  |
> | Data Aggregation (Poisson) | 705 MB     | 1219 MB |
> | Gaussian (d=64)            | 1239 MB    | 1239 MB |
> | MNIST (2,4,6)              | 4683 MB    | 6407 MB |
>
> - We note that in some of these experiments (e.g. Gaussian), peak GPU usage is due to the simulation and storage of the training samples for subsequent IMF steps, rather than during the network training. This can be reduced significantly by simulating in smaller batches, or by updating the cache during training rather than simulating all beforehand.
> - We also note that TreeDSBM has two major benefits over TreeDSB regarding run-time and memory requirements. Firstly, TreeDSB cannot be parallelised in the same way, as the training of the networks in each IPF iteration rotates around the edges sequentially. Secondly, the memory required to store training data for TreeDSB is far higher, as training a time-reversal along an edge requires saving entire trajectories simulated along the reverse direction, whereas TreeDSBM only requires storing endpoints (this is one of the most significant benefits of IMF over IPF).
>
> ## Application beyond barycentres
>
> Our initial motivation for our study was to utilise the IMF procedure in combination with the highly-performant fixed-point approaches for barycentre approximation. We include a detailed exposition of this perspective in the response to Reviewer 3rwT.
>
> We found that the tree-based framework provided the correct mathematical framework to combine these approaches in a simple and elegant way. Given our initial fixed-point motivation, the fact that our approach extends to more general tree settings was something of an added bonus that arose out of utilising the tree-based framework. Nevertheless, such problems have been utilised in works such as [5]. We conjecture that it could have applications in trajectory inference problems where populations might be known to 'split' according to a given structure (for example, maybe for single-cell dynamics). We unfortunately don't have the biological expertise to be confident in devising meaningful experiments for such settings, so instead provided a toy experiment in Appendix E.3 in which TreeDSBM appeared to work well, demonstrating fast and stable convergence. We have a brief paragraph discussing these wider problem settings in Appendix E.3, and **propose extending this to include the above discussion**, and **referencing this paragraph at the end of the Related Work section**. We hope future practitioners can build on our work for use in such settings.
>
>
> We thank the reviewer for their already kind feedback, and we hope that the contents of the Appendix and this rebuttal have addressed their comments. If you have any further questions, please let us know!
>
> ## References
>
> [1] Korotin et al, 2022, Wasserstein Iterative Networks for Barycenter Estimation
>
> [2] Korotin et al, 2021, Continuous Wasserstein-2 Barycenter Estimation without Minimax Optimization
>
> [3] Kolesov et al, 2024, Estimating Barycenters of Distributions with Neural Optimal Transport
>
> [4] Fan et al, 2021, Scalable Computations of Wasserstein Barycenter via Input Convex Neural Networks
>
> [5] Haasler et al, 2021, Multi-marginal Optimal Transport with a Tree-structured cost and the Schrödinger Bridge Problem

---

### Official Review · Reviewer_HGM9 · 2025-07-02

**Clarity:** 3
**Significance:** 2
**Originality:** 3
**Rating:** 4
**Confidence:** 3

**Summary:**

This paper extends the Iterative Markovian Fitting (IMF) procedure to solve tree-structured Schrödinger Bridge (TreeSB) problems, proposing the TreeDSBM algorithm. The work addresses a gap in the literature where only IPF-based methods (TreeDSB) existed for this setting, while IMF approaches have shown superior properties in the standard two-marginal case. The TreeSB problem generalizes optimal transport to multiple marginals connected by a tree structure, with Wasserstein barycenters as a special case when considering star-shaped trees. The authors demonstrate theoretical convergence guarantees and show empirical improvements over TreeDSB on synthetic 2D data, MNIST digits, and high-dimensional face datasets.

**Questions:**

Scalability concerns: How does the computational cost scale with tree size compared to TreeDSB? You mention parallel training of edges, but what about memory requirements for storing $2\times∣E∣$ networks and the overall wall-clock time comparison?

⁠Failure mode analysis: The CelebA experiment suggests the method may not be suitable for certain types of optimal transport problems. Can you provide more principled criteria for when TreeDSBM is expected to work well versus poorly? This would help practitioners understand when to apply your method.

⁠Convergence rate: While you show convergence, what can you say about the convergence rate compared to TreeDSB? The 6 vs 50 iterations comparison is compelling, but I'd like to see this validated across different problem settings and regularization parameters.

**Ethical Concerns:**

["NO or VERY MINOR ethics concerns only"]

**Final Justification:**

My concerns about Experimental validation have been addressed.  While questions remain about scalability, I believe the methodological contribution outweighs the practical limitations. Therefore I would like to increase the score.

**Limitations:**

Yes.

**Paper Formatting Concerns:**

NA.

**Quality:**

2

**Strengths And Weaknesses:**

Strengths:

In my opinion the paper’s main strength is a very clean theoretical generalization of IMF to multi‑marginal settings, together with a convergence proof of the TreeIMF. The resulting algorithm is embarrassingly parallel on each edges, and at least on the reported tests, run much faster than IPF‑style solvers.

Empirically, the results convincingly demonstrate the advantages of IMF over IPF in the tree setting. The 2D synthetic experiment shows TreeDSBM achieving comparable quality to TreeDSB with 6 IMF iterations versus 50 IPF iterations, which is a substantial computational improvement. The connection to fixed-point barycentre methods (Section 3.4) provides nice intuition for the star-shaped case.


Weaknesses:

My main concern is with the experimental validation, particularly the lack of rigorous quantitative evaluation on higher-dimensional problems. While the 2D case uses Sinkhorn divergence against a ground truth, the MNIST and CelebA experiments rely primarily on visual assessment. For a method claiming to advance the state-of-the-art, I would expect more comprehensive benchmarking.
The CelebA experiment reveals a significant limitation that I think deserves more critical examination. The authors acknowledge that their samples "do not match the visual quality of state-of-the-art results" and hypothesize this is due to the iterative nature being unsuitable for color alteration tasks. However, this raises questions about the general applicability of the approach - if the method struggles with an important class of transport problems, this significantly limits its practical utility.
I'm also skeptical about some implementation details. The method requires learning 2∣E∣2|E|
2∣E∣ vector fields (forward and backward for each edge), which could become computationally expensive for large trees. The authors mention parallelization but don't provide concrete scalability analysis or comparison of training times with TreeDSB.

The theoretical analysis, while correct, relies heavily on "mild assumptions" that are relegated to the appendix. For a theory-focused paper, I would prefer seeing the key assumptions stated explicitly in the main text, particularly since the tree setting introduces additional complexity compared to standard IMF.

---

> ### Author Rebuttal · Authors · 2025-07-30
>
> We thank the reviewer for their feedback, and we address their comments and questions below. We first provide a **brief rebuttal summary**:
> - **Experimental validation**: There are several further experiments in the Appendix, many of which address concerns of the reviewer. In particular, there are comparisons with alternative barycentre algorithms, and discussion of runtimes. If accepted, we intend to use the additional space to move some of these to the main paper.
> - **Scalability**: We address the reviewer's concerns in turn below. In particular, in the majority of use-cases, network sizes will be fairly small and thus not pose a computational bottleneck; in cases where it does, we outline ways it can be improved.
> - **CelebA**: We have subsequently improved performance on this benchmark by using a shared architecture across the edges.
>
> We expand on these points in more detail below, and highlight any proposed changes for revisions in **bold**.
>
> ## Experimental validation
>
> We note that there are several further experiments in the Appendix (Sections D and E), which could not be included in the main paper submission due to space constraints. We anticipate the reviewer may not have seen these, based on their comments. We hope these results allay the reviewer’s main concern regarding experimental validation.
>
> In these experiments, we include comparisons with state-of-the-art methods WIN [1], W2CB [2] and NOTWB [3]. These methods are chosen to be representative of different approaches in the literature— WIN is an iterative fixed-point approach, W2CB is an Input Convex Neural Network-based method, and NOTWB is based on recent Neural OT methodology.
>
> In particular, we include in Appendix E comparisons on a subset posterior aggregation experiment (a standard experiment in the continuous Wasserstein-2 barycentre literature, e.g. in [2],[4]), and also on high-dimensional Gaussian examples for $d=64,96,128$ (also a standard experiment, e.g. in [1],[2],[3]). In lines 1130-1145 we provide a discussion regarding the findings, highlighting the pros and cons of different approaches in the literature.
>
> Importantly, **we intend to move the subset posterior aggregation and high-dimensional Gaussian experiments to the main paper** if accepted.
>
> In Appendix D there are also more detailed examinations of the experiments in the main paper, including runtime analysis, barycentre progression during IMF iterations, and the effect of $\varepsilon$.
> - Run-time analysis: We discuss run-times of the various algorithms in the Appendix in lines 941-952, and lines 1079-1095, and TreeDSBM appeared favourable compared to existing barycentre solvers.
> - Ablations: We will add additional results concerning the role of batch size, and regularisation value $\varepsilon$ to the Appendix—please see response to reviewer QVwU.
>
> ## Implementation details and scalability
>
> Number of edges:
> - The reviewer notes a concern that learning $2 |E|$ separate networks may cause scalability problems. We note that this is similar in many continuous Wasserstein barycentre algorithms, including TreeDSB and e.g. [1],[2],[3]. Fortunately, there are ways to improve this if necessary, and we also do not envisage this being a limiting factor for the vast majority of use cases.
> - As discussed in Appendix C, one can use a single network along each edge and additionally condition on the direction. As the reversed drifts should be approximately the negative of each other, this can be efficiently parameterised as $(-1)^{dir} v_{\theta}(x,t,dir)$ (as done in [6], who show it has minimal impact on performance).
> - More generally, one could use a single shared network by additionally conditioning on the edge $e$. This makes particular sense for problems with shared structure between edges.
> - In our experiments, memory bottlenecks only arose in the CelebA experiment, which is exactly a scenario in which there is shared structure. We have subsequently used this single-network parameterisation and found improved performance (see below).
> - We propose to **extend the discussion in Appendix C** to include conditioning on edges too.
>
> Memory requirements, wall-clock time
> - Our algorithm offers a trade-off between memory consumption and run time due to its parallel nature. Namely, one can train edges simultaneously if compute allows (on a single GPU if enough memory, or parallelised across GPUs). If this cannot be done, one can train sequentially instead (in which case memory requirements for training would be comparable with standard bridge-matching).
> - In our experiments, we were (other than for CelebA) able to train simultaneously on a single GPU as the networks are not that large. Please see the response to reviewer ZjRa for further information regarding GPU usage. We anticipate that the majority of use-cases in practice would be for pointcloud data (e.g. the data aggregation experiment), which would use small MLP-like architectures with low memory requirements.
> - Comparison to TreeDSB: We note that TreeDSBM has two major benefits over TreeDSB regarding run-time and memory requirements. Firstly, TreeDSB cannot be parallelised in the same way, as each IPF iteration rotates the training around the edges sequentially. Secondly, TreeDSB requires saving entire trajectories to train the time-reversals, while TreeDSBM only requires storing the endpoints (this is one of the most significant benefits of IMF over IPF).
>
> ## CelebA
>
> We currently provide an extended discussion regarding the CelebA experiment in lines 1006-1029 in the Appendix.  In particular, we identify that the initial coupling (using bridge-matching models along each edge) does not always provided images that have structural similarity, so the resulting initial barycentre samples sometimes consist of distinct overlaid images. The subsequent IMF steps struggle to recover from this, resulting in the artifacts visible in some of the generations.
>
> In this benchmark, the true maps are relatively simple (primarily just colour changes), but TreeDSBM struggles to recognise this simpler structure due to infrequent communication between edges (which occurs only after each IMF iteration). In contrast, methods which perform well on this benchmark optimise all the maps together and with much more interaction between them, providing more scope for identifying the shared structure present in this specific problem. We hope this helps clarify for practitioners when TreeDSBM is likely to perform well or poorly.
>
> We hypothesised in the Appendix that different architectures could make TreeDSBM more competitive in this setting. We have since used a shared architecture as outlined in the previous section, which has improved the results (though they remain short of the best-performing methods for this problem). By learning features jointly across edges, the bridge-matching models behave more similarly along each edge, yielding coupling samples with greater structural similarity.
>
> We thus propose to **include in the Appendix a comparison of the results for the two parameterisations, to illustrate the role that architecture can play** in such highly-structured settings. We emphasise that much of the behaviour discussed here is likely specific to this particular benchmark; we anticipate the majority of practical use-cases would be for pointcloud data, for which similar behaviour would not arise.
>
>
> ## Convergence Rate
>
> We are glad that you find the fast convergence of TreeDSBM compelling. As discussed in the response to reviewer 3rwT, we believe that the TreeDSBM algorithm inherits good convergence properties from the similar fixed-point barycentre algorithms, which are known to converge quickly (see e.g. [7], who show good results with just a single iteration).
>
> We can include quantitative results regarding convergence speed in terms of IMF iterations in the Appendix. We provide such results for the 2d and data aggregation experiments below.
>
> | IMF step |1|2|3|4|5|6
> | - | - | - | - | - | - | - |
> | 2d, sinkhorn divergence $\downarrow$ |17.2 |1.27 |1.19  |1.14 | 1.07 | 1.04 |
> | Data Agg. (Poisson), BW2-UVP $\downarrow$| 31.1 | 0.010 | 0.011  | 0.012
>
>
> We can also provide similar results for TreeDSB. Note this would be expected to converge much slower, particularly for small epsilon, as fitting of the marginals only arises through iteratively learning time-reversals— if the forward processes rarely reach the other marginals (as is the case for smaller epsilon) this is very slow.
>
> ## Theoretical assumptions
>
> The key assumptions for the Brownian case are that $\int \lVert x \rVert^2 d \mu_i < \infty$ and $H(\mu_i)< \infty$ for each marginal $i$ (see proof of Theorem 3.1). We will **mention these in the main paper**. More generally, the required assumptions are stated in Proposition B.3 (for a general reference measure $\mathbb{Q}$), as well as standard technical assumptions inherited from the DSBM paper [8, Appendix C.1].
>
> We hope that the presence of experiments in the Appendix that are relevant to many of your questions, along with our comments and results above, and our comments in other rebuttals, have addressed your concerns. If you have any further questions, please let us know!
>
> ## References
>
> [1] Korotin et al, 2022, Wasserstein Iterative Networks for Barycenter Estimation
>
> [2] Korotin et al, 2021, Continuous Wasserstein-2 Barycenter Estimation without Minimax Optimization
>
> [3] Kolesov et al, 2024, Estimating Barycenters of Distributions with Neural Optimal Transport
>
> [4] Fan et al, 2021, Scalable Computations of Wasserstein Barycenter via Input Convex Neural Networks
>
> [5] Haasler et al, 2021, Multi-marginal Optimal Transport with a Tree-structured cost and the Schrödinger Bridge Problem
>
> [6] De Bortoli et al, 2024, Schrödinger Bridge Flow for Unpaired Data Translation
>
> [7] von Lindheim, 2022, Simple Approximative Algorithms for Free-Support Wasserstein Barycenters
>
> [8] Shi et al, 2023, Diffusion Schrödinger Bridge Matching

---

> > ### Comment · Reviewer_HGM9 · 2025-08-05
> >
> > Thanks to the authors for clearing most of the raised concerns. I would raise the score.

---

### Official Review · Reviewer_QVwU · 2025-07-02

**Clarity:** 3
**Significance:** 3
**Originality:** 3
**Rating:** 4
**Confidence:** 3

**Summary:**

This paper extends the standard iterative Markovian fitting approach to tree structured problems. One important feature of this is that it covers Wasserstein barycentres, gives rise to interesting applications.

**Questions:**

1) I think introduction overly focuses on technical aspects and relevant methods, but leaves a question mark for the nonspecialist reader why it is even of interest to look at these problems. Could you please provide a paragraph, aiming at an average NeurIPS reader, why your paper is of interest to this community?

2) Including above - why is the tree setting of special interest beyond barycenters?

3) I think, again, for someone who is not familiar with the very specialist earlier work, it is hard to understand all abstract definitions provided in this paper. Can authors provide a simple example, e.g., for a Gaussian setting (or whatever is appropriate) and make the problem fully explicit? This will make everything concrete for a reader who is not already working on SDEs on trees. Otherwise, the paper risks having an audience that is too narrow. It would be great if TreeDSBM (Algorithm 1) can be made explicit in a simple setting as an example.

4) Can authors elaborate a bit the challenges and observations about the training process? How does the cost scale with dimension or number of data points? Is there any instability issues? These would be good to mention in limitations if any. It would be better to clearly demonstrate the stability claims for your method by comparing it to relevant adversarial methods - and demonstrate this point specifically.

5) Is there any guideline to choose $\varepsilon$ for your method - which seems to impact the results?

6) Many hyper-parameters (training steps, batch size, network width, $\varepsilon$) vary across experiments; no ablation on drift-network architecture, number of edges trained, or tolerance to poor initial couplings. Demonstrating failure cases better would give a better picture.

7) How do introduced neural network architectures play with the assumptions required for theory? Please discuss.

8) Is it at all possible to generalize your framework beyond quadratic costs and non-Brownian cases?

**Ethical Concerns:**

["NO or VERY MINOR ethics concerns only"]

**Final Justification:**

My concerns are addressed but in general I think the paper's impact is limited and still too technical (the changes requested was a bit hard during a rebuttal period anyway). Thus I'd like to keep my score.

**Limitations:**

Yes

**Quality:**

4

**Strengths And Weaknesses:**

Strengths: There are multiple strengths of this work - notably

1) The paper is very well written from a technical perspective. Mathematical structure is made very clear throughout, clear literature review with connections to IPF and other ideas.
2) The paper goes beyond simplistic experiments, with MNIST and image experiments.

Weaknesses: While mathematically clean and well-written, the paper risks being obscure to most of the NeurIPS readers, see some suggestions to improve clarity.

---

> ### Author Rebuttal · Authors · 2025-07-30
>
> We thank the reviewer for their positive and constructive comments, and we address their comments and questions below. We first provide a **brief rebuttal summary**:
> - 'Overly technical presentation': We propose to:
>     - add a paragraph to the introduction regarding the importance of the Wasserstein barycentre problem.
>     - add a paragraph to the introduction emphasising the fixed-point interpretation, which we anticipate may be more intuitive for a reader less familiar with related works.
>     - add an explicit description of the algorithm in the simplest $\frac{1}{K}$-barycentre case.
> - 'Challenges in training/hyperparameters': We address each of these in turn below, with reference to relevant experiments in the Appendix, and with new ablations on the role of batch size and $\varepsilon$.
>
> We expand on these points and the reviewer's questions in more detail below, and highlight any proposed changes for revisions in **bold**.
>
> ## "Overly technical presentation"
>
> We thank the reader for highlighting this, as (behind the mathematical notation) we believe our algorithm is in fact quite intuitive, so are keen to convey this to the reader.
>
> ### "why it is even of interest to look at these problems"
>
> We propose to **add a paragraph in the introduction explaining the importance of the Wasserstein-2 barycentre problem**. In particular, we will highlight its interpretation as the natural notion of ‘mean’ for probability distributions, and its applications in Bayesian learning, clustering, and representation learning to name a few.
>
> ### "overly focuses on technical aspects and relevant methods", "Can authors provide a simple example"
>
> Currently the narrative of the paper focuses on TreeDSBM being an IMF counterpart to TreeDSB, which we agree may be somewhat opaque for a reader unfamiliar with these works. Our approach was in fact initially motivated by trying to combine IMF with fixed-point approaches for barycentre computation (please see our response to Reviewer 3rwT). We believe that the fixed-point perspective — namely that each point in the 'average' measure, is the 'average' of its corresponding points in the marginals — may be more intuitive for a reader less familiar with related work.
>
> While discussed in the paper, we believe a less familiar reader would benefit from having this interpretation highlighed more prominently. We thus **propose to add a paragraph in the introduction highlighting this motivation and interpretation**, and to **add a fully explicit description of Algorithm 1 in the simplified case of barycentres with weights 1/K**. This is the simplest setting (as the barycentre is constructed via $Y_0$ centred at $\sum_i \frac{1}{K} Y_i$, i.e. the ‘average’ of its corresponding points in the marginals), and should be intuitive for the reader.
>
> ### "why is the tree setting of special interest beyond barycenters"
>
> Given our initial fixed-point motivation described above, the fact our approach extends to the tree setting was something of an added bonus that arose out of using the tree-based mathematical framework. Nevertheless, such problems have been utilised in works such as [1]. We conjecture that it could have applications in trajectory inference problems where populations might be known to 'split' according to a given structure (for example, maybe for single-cell dynamics). We unfortunately don't have the biological expertise to be confident in devising meaningful experiments for such settings, so instead provided a toy experiment in Appendix E.3 in which TreeDSBM appeared to work well, demonstrating fast and stable convergence. We have a brief paragraph discussing these wider problem settings in Appendix E.3, and **propose extending this to include the above discussion**, and **referencing this paragraph at the end of the Related Work section**. We hope future practitioners can build on our work for use in such settings.
>
> ## Challenges in training/hyperparameters
>
> Scaling with number of datapoints/dim:
> - As it uses bridge-matching (trained via gradient descent on minibatches), our approach inherits the same scalability with datapoints as other flow-matching style algorithms.
> - In terms of dimension, we included experiments for high-dimensional Gaussians in Appendix E.2 for $d=64, 96, 128$, and observed no significant performance drop as $d$ increased.
>
> Instability:
> - We did not encounter instability issues, which we attribute to using stable bridge-matching losses, and an iterative procedure rather than adversarial losses.
> - "demonstrate the stability claims": In the Appendix, we have also compared to a range of state-of-the-art solvers [2],[3],[4]. Regarding stability, we found the solvers [4] (adversarial loss) and [3] (single regularised loss) to fail on the 2d-experiment. We discuss such issues in lines 935-940 in the Appendix.
> - We propose **adding a sentence referencing this in the main body**.
>
> Error in marginals:
> - We believe the main algorithmic limitation is that errors can accumulate in the marginals as IMF iterations proceed; this is a limitation inherited from standard IMF. We discuss this in lines 249-251 in the main paper, and in lines 851-858 in the Appendix. Rotating the starting marginal, as in [5], helps mitigate this.
> - It is therefore important that the bridge-matching steps fit the marginals accurately, which is affected by the hyperparameters training steps, batch size, network width. The effect of these parameters on bridge-matching is well-understood and they act in the expected ways. To provide an indication of how the learned bridge-matching quality affects the overall solution, we provide an ablation for batch size below on the 2d and data aggregation experiments for the hyperparameters used in the paper (similar effects occur for increasing training steps).
>
> | Batch size                                      | 64       | 256     | 1024   | 4096   |
> | - | - | - | - | - |
> | 2d, sinkhorn divergence $\downarrow$            | 1.57     | 1.24    | 1.04   | 1.04   |
> | Data Aggregation (Poisson), BW2-UVP $\downarrow$| 0.032    | 0.017   | 0.013  | 0.012  |
>
> - Network width: any sufficiently-expressive network will work similarly well. Fairly small MLP networks are sufficient for the pointcloud data experiments (i.e. 2$d$, subset aggregation, high-dim Gaussians).
> - We propose to **extend the discussion in Appendix lines 851-858**, by incorporating the above discussion and ablations.
>
> Role of $\varepsilon$
> - This is a perennial question in entropic OT. Standard methods to choose $\varepsilon$ in commonly used OT libraries (e.g. POT, OTT-JAX—done in proportion to the costs) provide good guidance for choosing suitable values. We add a few additional comments below.
> - Typically, one may want to use as small an epsilon as possible (for less entropic bias). One advantage of TreeDSBM over TreeDSB is that it allows for much smaller epsilon (TreeDSB does not converge for too-small epsilon as simulated trajectories do not reach the other marginals).
> - There is a visualisation of the role of epsilon in Figure 5 in the Appendix, and also for two values of epsilon for the MNIST experiment in Figure 7. We propose to **add quantitative results for varying epsilon to the Appendix**. Results for the 2d and data aggregation experiments are shown below.
>
> | $\varepsilon$                                   | 1.0      | 0.3     | 0.1   |
> | - | -| - | - |
> | 2d, sinkhorn divergence $\downarrow$            | 1.24     | 0.99    | 1.02   |
> |                                                 | 1e-3     | 3e-4    | 1e-4   |
> | Data Aggregation (Poisson), BW2-UVP $\downarrow$| 0.012    | 0.008   | 0.008  |
>
> ## Neural Architectures and Assumptions
>
> The theoretical analysis requires the SDE drifts to be locally Lipschitz, which is inherited from DSBM (for which one reason is to make use of known results regarding the well-posedness of the Doob’s $h$-transform). This is a weak regularity requirement, and is satisfied by standard neural architectures.
>
>
> ## Beyond Quadratic Costs, non-Brownian Case
>
> In its current form, the algorithm works only for the quadratic cost due to the Brownian reference measure (as with DSBM). We provide some thoughts regarding this below.
>
> - While DSBM works only for the quadratic cost and $\sigma>0$, [6] shows that ReFlow (i.e. $\sigma=0$) converges to OT for a convex cost $c(x,y) = h(x-y)$ if one restricts the form of the drift, and uses a Bregman divergence loss. One could potentially replace the bridge-matching steps with this procedure. The barycentre update would have to be modified accordingly (likely from $\sum_i \lambda_i y_i$ to $argmin_z \sum_i c(z, y_i)$ [7], though this may not be available in closed form).
> - Extensions of IMF to more general settings exist (e.g. state-dependent costs [8]). Incorporating such ideas into our framework is a promising direction.
> - Non-Brownian reference measures: as with DSBM, our method supports any reference process for which conditional bridges can be sampled (e.g. linear drift processes, see Appendix B of [5]). We focus on the Brownian case in the main paper due to its connection to OT.
>
>
> We hope that these comments have addressed your questions and concerns. If you have any further questions, please let us know!
>
> ## References
>
> [1] Haasler et al, 2021, Multi-marginal Optimal Transport with a Tree-structured cost and the Schrödinger Bridge Problem
>
> [2] Korotin et al, 2022, Wasserstein Iterative Networks for Barycenter Estimation
>
> [3] Korotin et al, 2021, Continuous Wasserstein-2 Barycenter Estimation without Minimax Optimization
>
> [4] Kolesov et al, 2024, Estimating Barycenters of Distributions with Neural Optimal Transport
>
> [5] Shi et al, 2023, Diffusion Schrödinger Bridge Matching
>
> [6] Liu, 2022, Rectified Flow: A Marginal Preserving Approach to Optimal Transport
>
> [7] Tanguy et al, 2025, Computing Barycentres of Measures for Generic Transport Costs
>
> [8] Liu et al, 2023, Generalized Schrödinger Bridge Matching

---

> > ### Comment · Reviewer_QVwU · 2025-08-03
> >
> > Thank you for answering my questions. I have no further questions.

---

### Official Review · Reviewer_3rwT · 2025-07-02

**Clarity:** 3
**Significance:** 3
**Originality:** 2
**Rating:** 4
**Confidence:** 3

**Summary:**

This paper adapts an existing technique to estimate Schrödinger bridges between two distributions---the so-called Iterative Markovian Fitting (IMF)---to the case of more-than-two distribution (multi-marginal OT) with tree structured cost (of which a typical example is star-shaped cost, encoding the celebrated Wasserstein barycenter problem).

The work proposes a set of precise definition to adapt the IMF (so far only introduced in the context of two-marginals) to this more general setting.

Markovian projection (one of the two steps to set up the IMF) _on a tree_ can be decomposed into performing Markovian projection on each edge, each of them being implemented as training (in a Monte-Carlo way) a NN to minimize a quadratic loss. This yield a (reasonably) tractable implementation of that scheme in practice.

Numerical experiments are conducted to showcase the feasibility of the method.

**Questions:**

As said in the Weakness section, I may be misunderstanding the numerical results showcased in Figure 3. May you elaborate a bit on this?

**Ethical Concerns:**

["NO or VERY MINOR ethics concerns only"]

**Final Justification:**

After discussion with the authors, I stick with my positive rating. I believe that this is a competent paper, and the authors' rebuttal clarified my potential concerns.

Because of its somewhat "incremental" (in that it mixes known techniques, even though effort is required to do so), I would not rank it as a top-tier paper, but I still believe that it may belong to NeurIPS and keep my positive rating (If I could, I would probably have rated it 4.5).

**Quality:**

3

**Strengths And Weaknesses:**

# Strengths

**Clarity:** The paper is clear and well written.

**Significance:** I believe that this approach is promising, it offers new perspective on dynamic entropic multi-marginal OT and reduce the gap between implementation and the underneath theoretical objects.

**Quality:** The paper is theoretically sound as far as I can tell.

# Weaknesses

**Originality:** One could perhaps argue that this approach is only mixing already established techniques. I nonetheless believe that this "had to be done" and would not consider this as being a strong limitation for publication.

**Quality:** On the numerical side, I am somewhat surprised with the results of Figure 3., which looks of much lower quality than those of https://arxiv.org/pdf/2402.03828 (cited in the work) for instance. Am I misunderstanding something?

---

> ### Author Rebuttal · Authors · 2025-07-30
>
> We thank the reviewer for their positive and constructive comments, and we address their comments and questions below. We first provide a **brief rebuttal summary**:
> - **'Originality'**: We propose to add a paragraph to the introduction highlighting the fixed-point barycentre interpretation more prominently, to highlight that our approach occupies an important position in the literature.
> - **'Figure 3'**: The noise in the samples in Figure 3 is expected, due to the entropy regularisation. There are ways to reduce/avoid this - see Figure 7 in the Appendix.
>
> We expand on these points in more detail below, and highlight any proposed changes for revisions in **bold**.
>
> ## "One could perhaps argue that this approach is only mixing already established techniques"
>
> We appreciate the reviewer for raising this point (and for also noting this is not a strong limitation). Currently the narrative of the paper focuses on TreeDSBM being an IMF counterpart to TreeDSB. We discuss below a slightly alternative perspective (which in fact initially motivated our approach), which we hope highlights more clearly why we believe our method occupies an important and prominent position in the literature.
>
> One of the most successful and elegant approaches for Wasserstein-2 barycentre computation is the iterative fixed-point approach, which involves iteratively updating a candidate barycentre $\nu_j$, by solving for the OT map $T_i$ from $\mu_i$ to $\nu_j$, and updating $\nu_{j+1}$ via the construction $\sum_i \lambda_i Y_i$ (where the $Y_i$ come from the coupling induced by the maps $T_i$). This approach was popularised by the seminal paper [1], and has formed the basis of many algorithmic developments, including in machine learning in [2] (one of the current SOTA approaches). Indeed, it has strong performance, and has been observed to converge quickly in only a few iterations [3]. However, a downside is that it requires solving $K \cdot N$ complete OT problems, where $K$ is the number of marginals and $N$ is the number of outer iterations, which is expensive as solving even a single OT problem can be challenging.
>
> The IMF procedure [4] has recently gained popularity for solving the entropy-regularised OT problem, and proceeds by iteratively performing bridge matching on its induced couplings. It is therefore a prominent question whether IMF can be used in combination with a fixed-point approach for solving for Wasserstein-2 barycentres. A naive approach would be to insert IMF to solve for each OT subproblem, requiring $K \cdot N$ complete IMF problems, which would in turn be $K \cdot N \cdot M$ bridge matching problems (moreover, making this approach grounded in theory would be non-trivial, as the fixed-point method holds only for the unregularised OT problem).
>
> The key aspect of our approach is that, rather than nesting the fixed-point iterations and the IMF iterations, they can in fact be elegantly *combined into a single iterative procedure*, along with a well-understood theoretical grounding.  In particular, our approach shows that the expensive OT computations required for the fixed-point procedure (the main limitation of such approaches), can instead be switched out for inexpensive bridge-matching procedures. Each fixed-point iteration is therefore significantly cheaper, and empirically we found the TreeDSBM algorithm to retain the fast convergence of the fixed-point procedure in terms of the number of iterations required.
>
> It is for this reason that we believe TreeDSBM is an *extremely natural algorithm for Wasserstein-2 barycentre computation*, and thus occupies a prominent position in the literature. We consider the fact it is a simple and elegant combination of the tree-structured framework with IMF to be one of its main strengths.
>
> We believe that the paper would benefit from advertising this fixed-point perspective more prominently, as currently the emphasis focuses on being an IMF counterpart to TreeDSB. We thus **propose to add a paragraph in the introduction highlighting this motivation and interpretation.**
>
> ## results of Figure 3
>
> We currently provide a discussion regarding this point in Section D.2 of the Appendix (lines 978-985) and in Figure 7. We **propose adding a reference to this discussion to this in the main paper**. We recall the main points from this discussion below.
>
> The results plotted in Figure 3 use a fairly large value of entropy-regularisation parameter $\varepsilon=0.02$ (which was selected to be more comparable with TreeDSB, which requires a large $\varepsilon$ to converge). This means that there *should* be noise present in an accurately computed solution, which is why the results look as they are. Indeed, the results shown in Figure 3 resemble those for other entropy-regularised barycentre approaches (see e.g. results for EGBARY in Figure 5 of the paper [5] linked by the reviewer).
>
> Clearly this may not be desirable; fortunately this can be reduced/avoided in two ways (which we do in the Appendix):
> - Using a smaller $\varepsilon$ value (e.g. $\varepsilon=0.001$)
> - Generating samples as $\sum_i \lambda_i Y_i$, where $Y_i$ are samples from the coupling over the marginals induced by the learned transport maps.
>
> Samples using both these approaches are included in Figure 7 in the Appendix (note minor typo, caption of (a) should read $\varepsilon=0.02$ rather than $0.2$). These samples are much more comparable with those in the work [5] referenced by the reviewer.
>
> ## Other changes
>
> Finally, we highlight here some additional changes that we propose, following feedback from other reviewers.
> - Move the comparisons with other continuous barycentre solvers [2].[5],[6] to the main body (they are currently in Appendix E.1 and E.2 due to space constraints)
> - Additional ablations on the role of epsilon, batch size (see response to reviewer QVwU).
>
> We hope that these comments have addressed your questions and concerns. If you have any further questions, please let us know!
>
>
> ## References
>
> [1] Alvarez-Esteban et al, 2016, A fixed-point approach to barycenters in Wasserstein space
>
> [2] Korotin et al, 2022, Wasserstein Iterative Networks for Barycenter Estimation
>
> [3] von Lindheim, 2022, Simple Approximative Algorithms for Free-Support Wasserstein Barycenters
>
> [4] Shi et al, 2023, Diffusion Schrödinger Bridge Matching
>
> [5] Kolesov et al, 2024, Estimating Barycenters of Distributions with Neural Optimal Transport
>
> [6] Korotin et al, 2021, Continuous Wasserstein-2 Barycenter Estimation without Minimax Optimization

---

> > ### Comment · Reviewer_3rwT · 2025-08-01
> > **Thanks**
> >
> > Thank you for answering my review. I did not notice this discussion in the Appendix (which, indeed, is worth referring to in the main body to avoid confusion as mine!).
> >
> > I will engage in discussion with other reviewers.

---

> > > ### Author Response · Authors · 2025-08-08
> > >
> > > We are pleased to hear that our rebuttal has addressed your question, and we would like to thank the reviewer for their positive and helpful comments during the review process.
> > >
> > > Best,
> > >
> > > The Authors

---

### Decision · Program_Chairs · 2025-09-17

**Decision:**

Accept (poster)

**Comment:**

The authors consider a Schrodinger bridge problem. They solve its generalized variant, namely, multi-marginal case with tree-structured cost. Wasserstein barycenter problem is a particular case of such tree-structured cost. They generalized a well-known Iterative Markovian Fitting approach to solve this problem.
Experimental verification confirmed ability of the approach to solve the Schrodinger bridge problem for tree-structured cost.


As a strengths of the paper I would highlight
- generalization of IMF to solve a schrodinger bridge problem with tree-structured cost
- the text is well-written

As a weaknesses I would highlight
- somewhat weak empirical evaluation on multidimensional problems

The main reasons for the accept decision are
- generalization of a solution of the important problem to a more general and important case
- clear text and theoretical explanations
- interesting insights about why the approach is more efficient for classical Wasserstein barycenter problem (see discussion with reviewer 3rwT)


During the discussion with the reviewers several issues were raised that should be addressed in the final version of the paper. Here is the list of the main issues
- the topic of the paper is rather techincal, so some more intuitive discussions are needed (reviewer QVwU)
- add discussion of why the proposed approach is so important for classical Wasserstein barycenter problem (reviewer 3rwT)
- comments to questions of reviewers HGM9 and ZjRa should be extended and included in the appendix, especially those concerning experimental evaluation